# Plant death caused by inefficient induction of antiviral *R*-gene-mediated resistance may function as a suicidal population resistance mechanism

Derib A. Abebe [1], Sietske van Bentum[1,2], Machi Suzuki[1], Sugihiro Ando[1], Hideki Takahashi[1] & Shuhei Miyashita [1 ✉]

Land plant genomes carry tens to hundreds of *Resistance* (*R*) genes to combat pathogens. The induction of antiviral *R*-gene-mediated resistance often results in a hypersensitive response (HR), which is characterized by virus containment in the initially infected tissues and programmed cell death (PCD) of the infected cells. Alternatively, systemic HR (SHR) is sometimes observed in certain *R* gene–virus combinations, such that the virus systemically infects the plant and PCD induction follows the spread of infection, resulting in systemic plant death. SHR has been suggested to be the result of inefficient resistance induction; however, no quantitative comparison has been performed to support this hypothesis. In this study, we report that the average number of viral genomes that establish cell infection decreased by 28.7% and 12.7% upon HR induction by wild-type cucumber mosaic virus and SHR induction by a single-amino acid variant, respectively. These results suggest that a small decrease in the level of resistance induction can change an HR to an SHR. Although SHR appears to be a failure of resistance at the individual level, our simulations imply that suicidal individual death in SHR may function as an antiviral mechanism at the population level, by protecting neighboring uninfected kin plants.

[1] Graduate School of Agricultural Science, Tohoku University, Sendai, Japan. [2] Department of Biology, Utrecht University, Utrecht, the Netherlands. ✉email: shuhei.miyashita.d7@tohoku.ac.jp

and plant genomes carry a class of genes collectively called *Resistance* (*R*) genes, which consist of tens to hundreds of genes per genome. Each gene encodes a receptor protein with a nucleotide-binding (NB) domain and a leucine-rich repeat (LRR) domain and confers resistance against specific pathogens. These gene products, which are called R proteins, directly or indirectly recognize pathogen infection, and induce resistance against the pathogen to inhibit its propagation[1]. In many cases, an R protein recognizes a single protein of a pathogen, thereby conferring resistance for that specific pathogen. Land plant genomes have accumulated many *R* genes through gene duplication and random substitutions, with diverse recognition specificity due to subsequent selection. For example, *RPP8*, *HRT*, and *RCY1*, which are allelic genes found in three different ecotypes of *Arabidopsis thaliana*, confer resistance to different pathogens: the oomycete pathogen *Hyaloperonospora parasitica*, turnip crinkle virus (TCV), and cucumber mosaic virus (CMV), respectively[2,3]. Eighteen R proteins are known to induce resistance against plant viruses[4]. Most of the well-studied antiviral *R* genes induce a hypersensitive response (HR), which is characterized by virus containment within the infected leaves and programmed cell death (PCD) induction in the infected regions. PCD in HR is thought to participate in virus containment within infected leaves; however, several studies have suggested that PCD is neither required nor sufficient for virus containment. For example, systemic movement of CMV was suppressed even in the *dnd1* mutant of *A. thaliana*, which is deficient in PCD induction[5] and virus accumulation was observed outside of necrotic lesions after HR induction against tobacco mosaic virus and potato virus Y (PVY)[6,7]. Extreme resistance (ER) is a form of *R*-gene-mediated antiviral resistance in which virus accumulation is suppressed below the detection level in initially infected cells without PCD induction[8]. Overexpression of HR-inducing *R* genes caused ER to TCV and CMV, implying that ER is a result of enhanced resistance induction[2,9] such that viral accumulation is inhibited before the detection level is reached. An *R* gene that confers ER against PVY to potato induces HR-like necrosis when PVY coat protein

(CP), the elicitor, was transiently expressed via Agroinfiltration[10]. This result also supports the idea that PCD is neither required nor sufficient for virus containment. Thus, the benefit of *R*-gene-mediated PCD in antiviral resistance remains controversial.

Systemic necrosis is among the most severe symptoms observed after viral infection of plants, and is often thought to result from susceptible interactions. However, several genetic studies in various virus–plant pathosystems indicated that systemic necrosis is controlled by single incompletely dominant or completely dominant *R*-gene loci[11–16]; the PVY–potato pathosystem induces ER, HR, and systemic necrosis depending on the combination of PVY isolates and potato cultivars[17,18]. Molecular studies have also indicated that systemic necrosis is induced in at least some cases via similar mechanisms to those of HR. As observed in *R*-gene-dependent HR, induction of pathogenesis-related genes and accumulation of hydrogen peroxide ($H_2O_2$) and salicylic acid were observed in systemic necrosis[19–22]. Similarly, PCD induction and rapid accumulation of reactive oxygen species were demonstrated in systemic necrosis; and the involvement of a common signal transduction pathway in systemic necrosis induction and HR induction was suggested[23,24]. Michel et al. demonstrated that systemic veinal necrosis by a PVY necrotic isolate (PVY^N) in tobacco was abolished by mutations in the *NtTPN1* gene, a putative *R* gene[25]. The systemic necrosis observed in the abovementioned studies could be the result of inefficient resistance induction, such that *R* genes fail to restrict the systemic spread of the viruses, but induce PCD afterwards. Thus, systemic necrosis caused by the same mechanisms that drive HR is sometimes called systemic HR (SHR). However, to the best of our knowledge, there has been no direct evidence that resistance induction is less efficient in SHR than in HR, due to a lack of methods for quantifying resistance induction efficiency.

In this study, we used the CMV–*N. benthamiana* pathosystem to develop a method to quantify the efficiency of resistance induction by detecting an *R*-gene-mediated decrease in viral multiplicity of infection (MOI), i.e., the average number of viral genomes that establish host cell infections. Based on the results of our analysis, we discuss the roles of SHR and PCD in population-level antiviral resistance of land plants and the trajectory of antiviral *R* gene evolution.

## Results

**Characterization of SHR induction in CMV(Y)–*N. benthamiana*/*A. thaliana* pathosystems.** *RCY1*, an *R* gene of *Arabidopsis thaliana* ecotype C24, induces HR upon infection by CMV Y strain [CMV(Y)][3,26]. CMV has a tripartite positive-strand RNA genome consisting of RNA1, RNA2, and RNA3 (Fig. 1a). A CMV (Y) variant that carries the *CP* gene from a non-HR-inducing CMV B2 strain [CMV(B2)] instead of its original *CP* gene did not induce HR to *A. thaliana* C24, implying that CP is directly or indirectly recognized by RCY1[27]. Studies using chimeric constructs between *RCY1* and its allelic ortholog *RPP8* have shown that a chimeric *R* gene, *RPRPCY*, which has N-terminal coiled-coil and NB domains from *RPP8* and a C-terminal LRR domain from *RCY1*, can induce HR against CMV(Y) in *N. benthamiana*[28]. Inoculation of CMV(Y) to *RPRPCY*-transformant *N. benthamiana* [*N. benthamiana* (*R*+)] plant usually results in HR. In the current study, we obtained a CMV variant from upper uninoculated leaves of a *N. benthamiana* (*R*+) plant that showed mosaic and necrotic symptoms 2 weeks after inoculation of wild-type (WT) CMV(Y) (Fig. 2a). Sequence analysis showed that the variant had a T45M (Thr45 to Met) substitution in its CP (Fig. 2b). Inoculation of *N. benthamiana* (*R*+) with in vitro-transcribed RNA3 with T45M substitution in CP together with WT RNA1 and RNA2-YFP [RNA2 labeled with

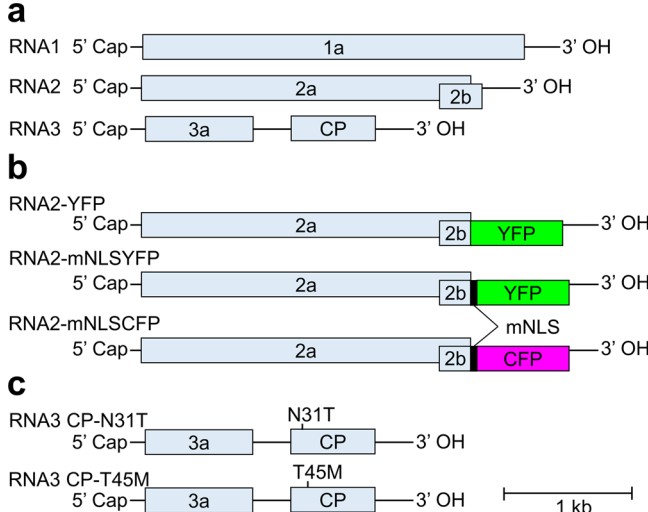

**Fig. 1 Genome structure of cucumber mosaic virus (CMV) and its variants used in this study. a** Genome structure of CMV. 1a and 2a proteins are replication proteins; 2b, an RNA-silencing suppressor; 3a, cell-to-cell movement protein; CP capsid protein. **b** RNA2 variants. Yellow and cyan fluorescent protein (YFP and CFP, respectively) were produced as fusion proteins to the N-terminal 2b protein with and without modified nuclear localization signal (mNLS) peptide. **c** RNA3 variants with 1-aa substitutions.

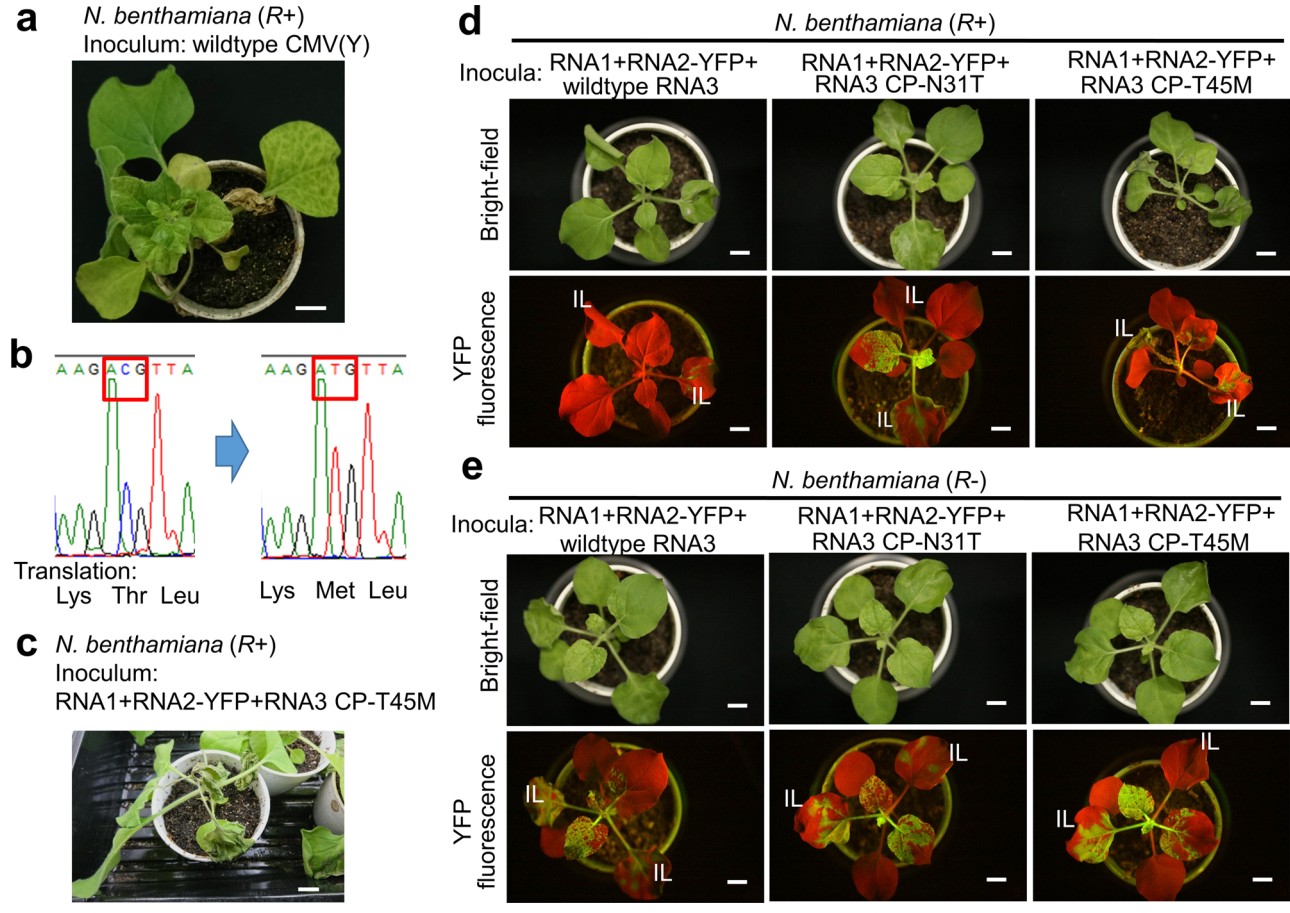

**Fig. 2 Characterization of a CP-T45M variant in *N. benthamiana*. a** Mosaic and necrotic symptoms appeared in upper uninoculated leaves of a *N. benthamiana* (*R*+) plant inoculated with wild-type (WT) CMV(Y) at 2 weeks after inoculation. **b** Sequence chromatograms for direct sequencing of reverse-transcription polymerase chain reaction (RT-PCR) product from WT and variant-infected plants. **c** An example of systemic necrosis observed at 1 week after inoculation of an RNA3 CP-T45M variant together with RNA1 and RNA2-YFP. **d**, **e** Representative images of bright-field and YFP-fluorescence observations of *N. benthamiana* (*R*+) plants d and *N. benthamiana* (*R*–) plants e at 4 days after inoculation of RNA1, RNA2-YFP, and WT RNA3 or CP variants. IL inoculated leaves. Note that uninfected regions of leaves appear red in YFP-fluorescence observations due to autofluorescence; infected regions appear yellow. Scale bars = 1 cm. Similar results were observed in more than three replicates of experiments.

**Table 1 Summary of observed phenotypes[a] after inoculations with different combinations of viral RNA3 genotypes and plant genotypes.**

| Inoculated RNA3 | *N. benthamiana* | | *A. thaliana* | | |
|---|---|---|---|---|---|
| | *R*(−) | *R*(+) | *R*(−) | *R*(+) | *R*(++) |
| Wildtype | sm | HR[b] | sm | HR | ER |
| CP-N31T | sm | sm | sm | sm | sm |
| CP-T45M | sm | SHR | sm | SHR | ER |

[a]*sm* systemic mosaic, *HR* hypersensitive response, *ER* extreme resistance, *SHR* systemic HR.
[b]SHR is observed at low frequency.

a *yellow fluorescent protein* (*YFP*) gene] reproduced systemic infection (Fig. 2c, d, right panels and Table 1 for summarized results). Of note, systemic necrosis was observed in all inoculated plants within 3–5 days after inoculation (Supplementary Table 1). Inoculation of WT *N. benthamiana* [*N. benthamiana* (*R*–)] with the same inoculum resulted in systemic infection without necrosis (Fig. 2e, right panels). Thus, systemic necrosis observed after inoculation of the T45M variant to *N. benthamiana* (*R*+) plants occurred in an *R*-gene-dependent manner; i.e., the systemic necrosis was SHR. Time-course observation showed that tissues with YFP fluorescence subsequently caused necrosis

(Supplementary Fig. 1), suggesting that this SHR necrosis occurred in infected tissues. Inoculation of WT RNA3 mixed with RNA1 and RNA2-YFP caused HR localized to inoculated leaves of *N. benthamiana* (*R*+) in 10 of 12 inoculated plants and caused SHR in two other plants at 5 or 6 days after inoculation (Supplementary Table 1), but caused systemic infection in *N. benthamiana* (*R*–) without necrosis (Fig. 2d, e, left panels). Another RNA3 variant that has an N31T (Asn31 to Thr) substitution in the CP and was previously found to escape *RCY1*-mediated resistance (Takahashi H., unpublished result) showed systemic infection in both *N. benthamiana* (*R*+) and *N. benthamiana* (*R*–) without necrosis (Fig. 2d, e, middle panels). A double mutant with N31T and T45M substitutions did not cause necrosis to *N. benthamiana* (*R*+) plants (Supplementary Fig. 2). This result further supports the idea that systemic necrosis observed after T45M variant inoculation to *N. benthamiana* (*R*+) plants was *R*-gene dependent. We confirmed that SHR tissues caused by the T45M variant and HR lesions caused by WT virus in *N. benthamiana* (*R*+) plants were similarly stained by 3,3'-Diamino-benzidine (DAB) (Supplementary Fig. 3a, b) and accumulated similarly fragmented DNA (Supplementary Fig. 3c, d), implying that the systemic necrosis in SHR was PCD dependent, as is the case for HR. Western blot analysis showed that viral CP cannot be detected in SHR or HR necrotic tissues, suggesting that viral particles are degraded upon cell death (Supplementary Fig. 4a).

This is consistent with our previous results in press-blot analysis that viral CP can be detected in the marginal regions of HR necrotic tissues but not within them in CMV–*A. thaliana* pathosystem[9]; the CP accumulation in the marginal regions of HR necrosis was also observed in CMV–*N. benthamiana* pathosystem used in this study (Supplementary Fig. 4d).

The genome of *A. thaliana* ecotype Col-0 does not carry *RCY1*. We used WT *A. thaliana* Col-0 [*A. thaliana* (*R*–)] and two transformants carrying different copy numbers of *RCY1* transgenes, one with a single copy of transgene [*A. thaliana* (*R*+)] and

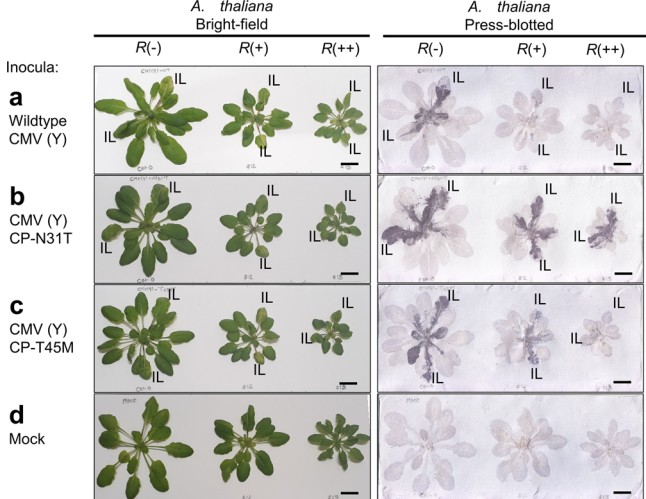

**Fig. 3 Characterization of a CP-T45M variant in *A. thaliana*.** Bright-field observations and viral CP detection by press-blot analysis at 7 days after inoculation of WT CMV(Y) (**a**) or its CP variants CMV(Y) CP-N31T (**b**) and CMV(Y) CP-T45M (**c**), or mock inoculation (**d**) to *A. thaliana* (*R*–), (*R*+), and (*R*++) plants. Dark areas in press-blot analysis results indicate CP accumulation, visualizing the virus distribution. IL inoculated leaves. Scale bars = 1 cm. Similar results were observed in more than three replicates of

another with ten copies [*A. thaliana* (*R*++)][9,29]. To detect systemic infection, a press-blot analysis of the inoculated *A. thaliana* plants was performed at 7 days after inoculation (Fig. 3), and occurrence of necrotic lesions or systemic necrosis was judged at 14 days after inoculation (Supplementary Fig. 5 and Table 1 for summarized results). As reported in Sekine et al.[9], inoculation of WT CMV(Y) resulted in HR and ER in *A. thaliana* (*R*+) and *A. thaliana* (*R*++), respectively, in contrast to systemic infection in *A. thaliana* (*R*–) (Fig. 3a). A CMV(Y) CP-N31T variant showed systemic infection in all three genotypes without necrosis, implying that the N31T variant escapes *RCY1*-mediated resistance (Fig. 3b). A CMV(Y) CP-T45M variant showed systemic susceptible infection in *A. thaliana* (*R*–), systemic infection followed by necrosis, thus SHR, in *A. thaliana* (*R*+), and ER in *A. thaliana* (*R*++) (Fig. 3c and Supplementary Fig. 5c). Consistent results were also obtained in a western blot analysis of upper uninoculated plant leaves (Supplementary Fig. 6). These results indirectly imply that the CP-T45M variant partially escapes *RCY1*-mediated resistance.

**MOI estimation in the presence and absence of the *R* gene.** Plant RNA viruses accumulate $10^4$–$10^7$ copies in each infected cell[30–35]. We previously showed that only a handful of copies of such a huge population can establish cell infection in adjacent cells after cell-to-cell movement, with MOIs of around 5 and 6 for the first and second cell-to-cell movements of Japanese soil-borne wheat mosaic virus in *Chenopodium quinoa*[36]. Similarly, the MOI of the first cell-to-cell movement of tomato mosaic virus in *N. tabacum* was around 4[37]. In the present study, we estimated the MOI of CMV(Y) and the abovementioned CP variants in the presence or absence of the *R* gene to assess the effect of *R*-gene-mediated resistance on viral MOI. To estimate MOI, we prepared CMV(Y) RNA2 variants that produce YFP or cyan fluorescent protein (CFP) as a fusion protein to the viral 2b protein. The addition of a modified nuclear localization signal (mNLS) peptide to the fluorescent proteins exhibited lower nuclear localization than the original NLS from the SV40 large T antigen, which allowed us to distinguish co-infected and singly infected cells. The

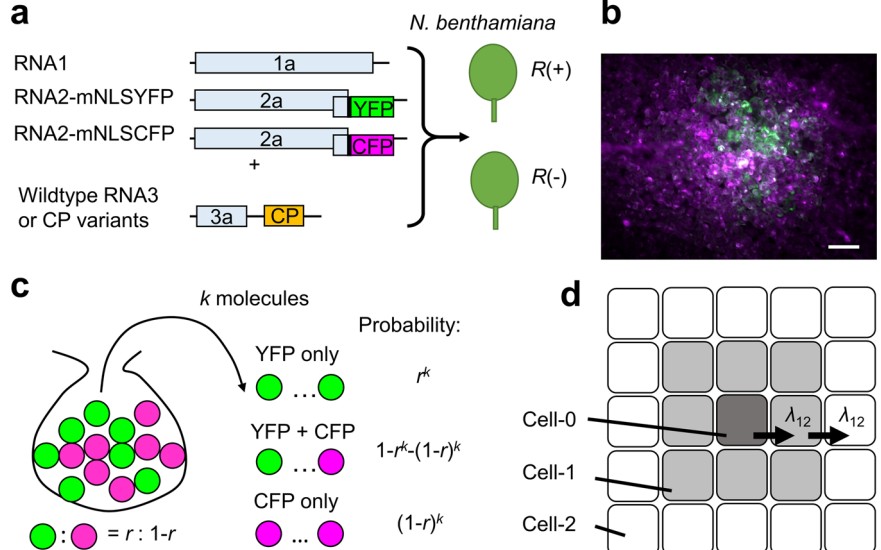

**Fig. 4 Multiplicity of infection (MOI) estimation in the presence and absence of the *R* gene. a** Schematic explanation of the inoculation procedures. **b** An example of stochastic separation of the two RNA2 variants at 20 h after inoculation to *N. benthamiana* (*R*–). YFP and CFP images were pseudocolored with green and magenta, respectively, and then merged. Scale bar = 100 μm. **c** Idea for MOI estimation: if a number of balls are randomly selected from a mixture of two types of balls, the probability that either of the two types is selected increases as the number of balls selected decreases. **d** Classification of infected cells by the number of cell-to-cell movements experienced by intracellular viral populations. MOIs for first and second cell-to-cell movements were assumed to be equal ($=\lambda_{12}$) in the current study.

**Table 2 Multiplicity of infection (MOI) estimates in wild-type (WT) and *RCY1*-transformant *Nicotiana benthamiana* plants inoculated with WT and variant RNA3s.**

| Inoculated RNA3 | MOI in *N. benthamiana* (R−) [A] | MOI in *N. benthamiana* (R+) [B] | *R*-gene-dependent MOI change [(B − A)/A] (%) |
|---|---|---|---|
| WT | $\lambda_{12} = 5.72 \pm 0.24$ | $\lambda_{12} = 4.08 \pm 0.22$ | −28.7 |
| CP-N31T | $\lambda_{12} = 5.75 \pm 0.36$ | $\lambda_{12} = 5.67 \pm 0.28$ | −1.4 |
| CP-T45M | $\lambda_{12} = 5.73 \pm 0.35$ | $\lambda_{12} = 5.00 \pm 0.26$ | −12.7 |

Mean ± standard error for each MOI estimate is shown.

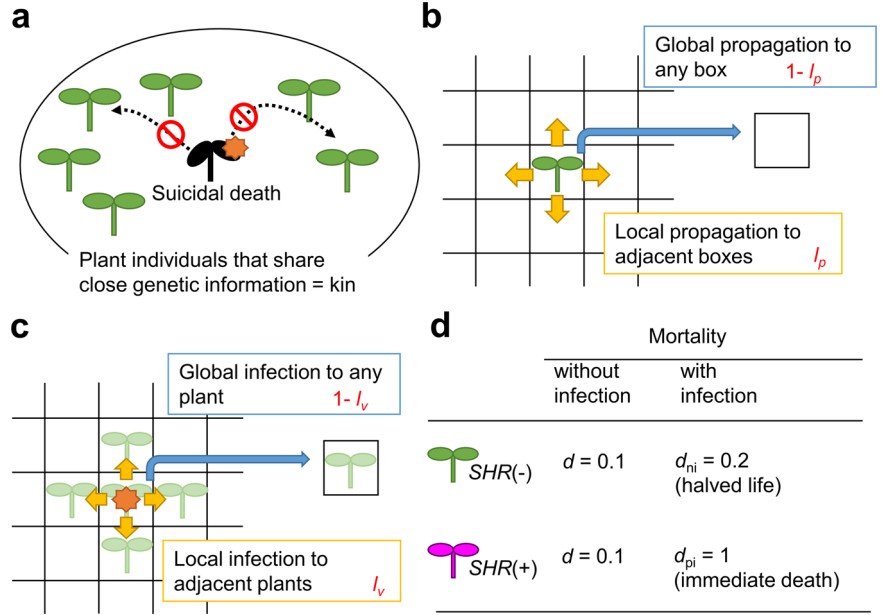

**Fig. 5 Conceptual model for a suicidal population resistance via SHR that may protect next of kin. a** Protection of adjacent kin plants by suicidal death. **b** Local and global propagation of plants. **c** Local and global propagation of virus. **d** Mortality of *SHR*(−) and *SHR*(+) plants with and without virus infection. Parameter values were set to answer the question: whether it is more fit to suffer lower reproduction due to viral infection, or to induce inefficient resistance against the virus, and, by opposing, end them.

RNA2 variants, named RNA2-mNLSYFP and RNA2-mNLSCFP, were mixed with WT RNA1 and WT RNA3 or its CP variants and inoculated to *N. benthamiana* (R−) and *N. benthamiana* (R+) (Fig. 4a). In theory, a small MOI enhances stochastic separation of the two RNA2 variants and increases the number of cells that are singly infected by either of the two fluorescent variants (Fig. 4b, c). Based on this idea, we estimated the MOI of RNA2 in each condition from the number of co-infected and singly infected cells; this process has been described in greater detail elsewhere[36,37]. Time-course observations showed that PCD was not observed until 24 h after inoculation to *N. benthamiana* (R+). At 14–16 h after inoculation, many of the infected sites showed 10–30 infected cells via YFP and/or CFP fluorescence. These sites were considered to have an initially infected cell (Cell-0), eight cells infected after the first cell-to-cell movement (Cell-1), and the remaining cells infected after the second cell-to-cell movement (Cell-2) (Fig. 4d). We counted the cells infected by both or either RNA2-mNLSYFP and RNA2-mNLSCFP in these sites and used the data for MOI estimation, assuming that the MOIs for the first and second cell-to-cell movements were equal (Supplementary Tables 2 and 3, raw data and Supplementary Text 1, R script used for MOI estimation).

The obtained MOI estimates for first and second cell-to-cell movements ($\lambda_{12}$) are summarized in Table 2. Inoculation of WT RNA3 resulted in $\lambda_{12} = 5.72 \pm 0.24$ (mean ± standard error) in *N. benthamiana* (R−), a compatible interaction, and $\lambda_{12} = 4.08 \pm 0.22$ in *N. benthamiana* (R+), an HR-inducing combination.

Thus, the *R*-gene-mediated decrease in MOI was 28.7%. Inoculation of the RNA3 CP-T45M variant exhibited modest decreases in MOI: $\lambda_{12} = 5.73 \pm 0.35$ in *N. benthamiana* (R−), a compatible interaction, and $\lambda_{12} = 5.00 \pm 0.26$ in *N. benthamiana* (R+), an SHR-inducing combination. The *R*-gene-mediated decrease in MOI was thus 12.7%. Such a decrease in MOI was not detected for the inoculation of the RNA3 N31T variant, which completely escaped plant resistance, further confirming that the MOI decreases under HR- and SHR-inducing conditions occurred in an *R*-gene-dependent manner. In summary, an *R*-gene-mediated decrease in MOI was observed under SHR-inducing conditions, but it was not as great as that under HR-inducing conditions, and the MOI decrease was detected prior to PCD induction under both HR- and SHR-inducing conditions.

**SHR as a possible resistance mechanism at the population level.** The above MOI estimations provide a direct evidence that resistance induction is less efficient in SHR than in HR. This finding strongly supports the idea that SHR occurs when antiviral resistance induction by an *R* gene is not efficient enough to initiate HR. Such a situation may occur frequently in nature because spontaneous viral mutation can weaken resistance induction by chance, as observed in the CMV(Y) CP-T45M variant. Furthermore, when an *R*-gene product happens to recognize a new viral protein during the diversification process, initial recognition intensity can often be weak and resistance induction can be inefficient, resulting in SHR. In both cases, SHR

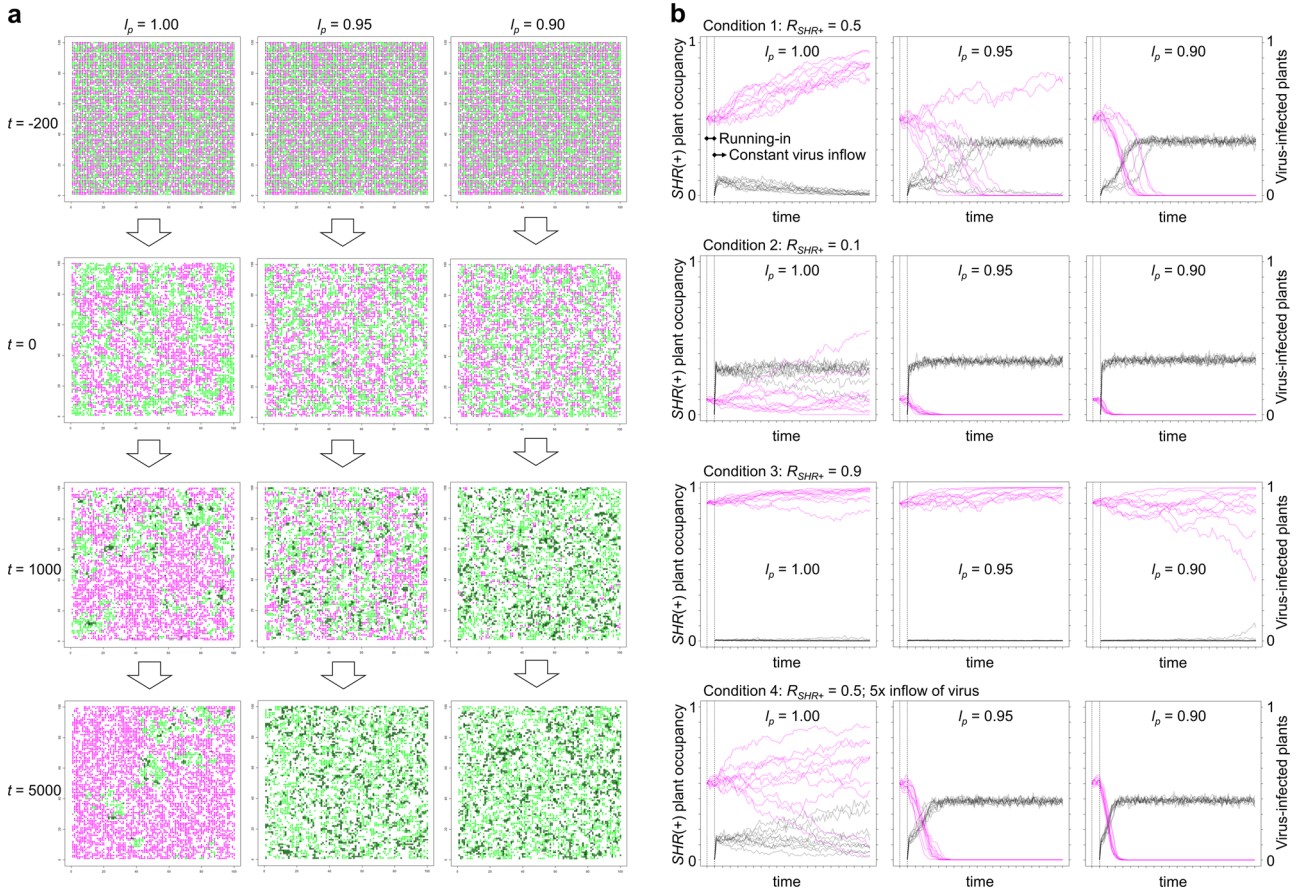

**Fig. 6 Simulation results for a suicidal resistance mechanism at different levels of dependency on local propagation ($l_p$). a** Time-course presentation of simulation results at different $l_p$ values under condition 1 ($R_{SHR+} = 0.5$). The results of single trials for each value of $l_p$ are shown. Green and magenta dots indicate $SHR(-)$ and $SHR(+)$ plants, respectively. The dependency on local infection of the virus was set to $l_v = 0.90$. After a short running-in period ($t = -200$ to $t = 0$), constant virus influx was started. Infected plants are shown as green or magenta dots surrounded by black squares. Magnified images can be found in Supplementary information file (Supplementary Figs 7–9). **b** Summary of 10 simulation trials for each condition. Magenta lines indicate $SHR(+)$ plants' occupancy in relation to the total number of plants. Black lines indicate the proportion of virus-infected plants among all plants.

kills the plant, which can never be more adaptive to the plant individual than full susceptibility (i.e., not having the *R* gene) especially since compatible viral infections rarely kill infected plants. From this perspective, early SHR-inducing *R* genes would not be selected and would rapidly disappear from the population. However, considering that land plants have diversified the recognition spectra of multiplicated *R* genes, the existence of SHR would need to be explained by one or more alternative mechanisms.

An alternative mechanism may be found in population-level antiviral resistance. Land plants often propagate via seeds and sometimes via vegetative reproduction. These mechanisms result in mainly local propagation of kin: plant individuals that share close genetic information generally also share spatial proximity. Viral transmissions from plant to plant also occur locally; many plant viruses are transmitted by aphid vectors, which typically fly no farther than 5 m[38]; some plant viruses are transmitted via soil, with or without the help of microorganisms, which are far less motile than aphids. From the perspectives of local plant propagation and local virus transmission, systemic death caused by SHR can help infected plants to avoid serving as a source of infection for adjacent kin plants, at the cost of their own lives (Fig. 5a).

To test whether and on what conditions having an SHR-inducing inefficient *R* gene can be more adaptive than being fully susceptible (i.e., without having the *R* gene), we developed a simple mathematical model to simulate plant and viral population dynamics in a two-dimensional $100 \times 100$-lattice boxes with periodic boundaries. Each box can be inhabited by one plant individual, which can deposit progeny in each of the four surrounding boxes at a probability of $r_p\,l_p/4$, and in any other (usually more distant) box at a probability of $r_p(1-l_p)/100^2$ in a unit of time, where $r_p$ is the total probability to produce progeny and $l_p$ parametrizes dependency on local propagation (Fig. 5b). The parameter $l_p$ changes the size of plant subpopulation patches in the absence of the virus (Fig. 6a, $t = 0$). Similarly, we assumed that viruses in an infected plant can spread to plant individuals in each of the four surrounding boxes at a probability of $r_v\,l_v/4$ and to any plants in a box within the lattice at a probability of $r_v(1-l_v)/100^2$ in a unit of time (Fig. 5c). We used $l_v = 0.90$ unless otherwise noted; this parameter was tested later. Two plant genotypes were included in the model: plants with an SHR-inducing *R* gene [$SHR(+)$ plants] and susceptible plants without the SHR-inducing gene [$SHR(-)$ plants]. Both plant types have the same mortality rate $d\ (= 0.1$; average life time $1/d = 10$) when they are not infected by the virus, but differ in mortality when infected. In $SHR(-)$ plants, $d_{ni} = 0.2$; after infection, the average life time decreases to $1/d_{ni} = 5$, and the expected number of progeny decreases accordingly. In $SHR(+)$ plants, $d_{pi} = 1$, indicating immediate death, with no progeny (Fig. 5d). Note that dead plants were not assumed to serve as infection sources anymore, because the above experiment showed that necrotic

tissues did not contain detectable amounts of the virus (Supplementary Fig. 4a). For simplicity, we assumed no changes in the reproductive rate due to viral infection; however, changes in average life time implicitly affect progeny numbers. We assumed constant influx of the virus to the lattice space after a short running-in period; we assumed no plant recovery from the virus or vertical transmission of the virus.

Simulations were performed for different $l_p$ parameter values to test the effect of strongly localized and more distant propagation on the survival of $SHR(+)$ plant populations. Initial simulations started with equal numbers of $SHR(+)$ and $SHR(-)$ plants (Fig. 6a, b, condition 1: $R_{SHR+} = 0.5$). At $l_p = 1.00$, when propagation occurs only very locally, $SHR(+)$ plants survived in all ten trials and gained population occupancy from 50 to 85.6% ± 6.2% (mean ± standard deviation) during the simulated period (Supplementary Table 4). At $l_p = 0.95$ and $l_p = 0.90$, $SHR(+)$ plants became extinct in eight and ten of ten trials in the same period, respectively. These results can be explained intuitively by recalling the ancient Greek phalanx: simulated $SHR(+)$ plant populations at $l_p = 1.00$ resemble a large phalanx, where only outside plant individuals are attacked by the virus; at $l_p = 0.95$ and $l_p = 0.90$, the phalanx is smaller, and therefore easily defeated (Supplementary Figs 7–9 for magnified images). Further simulations were started with an initial population of 90% $SHR(-)$ plants and 10% $SHR(+)$ plants to simulate a small subpopulation of plants obtaining an SHR-inducing $R$ gene by chance (Fig. 6b, condition 2: $R_{SHR+} = 0.1$). At $l_p = 0.95$ and $l_p = 0.90$, $SHR(+)$ plants rapidly became extinct in all ten trials, whereas at $l_p = 1.00$, only one of ten trials resulted in the extinction of $SHR(+)$ plants within the same time period; the occupancy of $SHR(+)$ plants decreased in six trials, but increased in four trials. These simulations imply that local reproduction helps $SHR(+)$ plants to survive for longer periods. This difference in survival time creates a much larger difference in the cumulative number of $SHR(+)$ plants before extinction, by orders of magnitude (Supplementary Table 5). Importantly, with a higher cumulative number of plants, $SHR(+)$ plants are more likely to obtain a genetic variant that can induce HR, for example, via improved pathogen recognition ability by amino acid substitutions in the $R$ gene or enhanced expression of the $R$ gene by promoter nucleic acid substitutions. Thus, even if $SHR(+)$ plant populations go extinct, the SHR trait can lengthen survival time and increase opportunities for adaptation if plants propagate in a highly localized manner. In condition 1, when simulations were started with equal numbers of $SHR(+)$ and $SHR(-)$ plants, $SHR(+)$ plants gained 1.7 times their population occupancy at $l_p = 1.00$, whereas no clear increase was observed with the same $l_p$ in condition 2, when simulations were started with 10% $SHR(+)$ plants. This result implies positive frequency-dependent selection for the SHR trait, which is explained by the abundance of virus-infected plants in the defined space; more $SHR(+)$ plants allow less virus within the space, resulting in a further increase in the abundance of $SHR(+)$ plants. Finally, simulations were performed with an initial population of 10% $SHR(-)$ plants and 90% $SHR(+)$ plants (Fig. 6b, condition 3: $R_{SHR+} = 0.9$). These conditions can occur when an SHR-inducing $R$ gene is lost from a subpopulation or when genetic drift caused the accumulation of newly occurring $SHR(+)$ plants by chance. Under these starting conditions, the virus population was kept small throughout the simulations, with $l_p = 1.00$, 0.95, and 0.90, implying that virus spread would be suppressed at the population level. The average occupancy of $SHR(+)$ plants showed no clear change at $l_p = 0.90$, from 90 to 85.7% ± 17.9%, but clearly increased at $l_p = 0.95$ and 1.00, from 90 to 97.6% ± 3.2% and 96.3% ± 4.9%, respectively (Fig. 6b, condition 3: $R_{SHR+} = 0.9$; Supplementary Table 4). This result implies that a low but constant influx of virus can function as positive selection

pressure for an $SHR(+)$ trait under local propagation conditions. When an enhanced influx of viruses was tested in the simulation model, $SHR(+)$ plants showed lower occupancy at $l_p = 1.00$ and rapid extinction at $l_p = 0.95$ and $l_p = 0.90$ (Fig. 6b, condition 4: $R_{SHR+} = 0.5$; 5× influx of virus; Supplementary Table 4), implying that a suicidal population resistance is not always evolutionarily successful, even at $l_p = 1.00$. Similarly, less local transmission of the virus ($l_v = 0.80$ and 0.70) resulted in the extinction of $SHR(+)$ plants at $l_p = 1.00$, 0.95, and 0.90 (Supplementary Fig. 10). Assumption of one-time virus introduction instead of constant influx resulted in increase in occupancy of $SHR(+)$ plants at $l_p = 1.00$, when the simulation was started with an initial population of 50% $SHR(+)$ plants; however, when simulations were started with an initial population of 90% $SHR(+)$ plants, the increase was stopped after a virus extinction due to the lack of selection pressure (Supplementary Fig. 11). Such a situation can be also observed in nature, not only for SHR-inducing $R$ genes but also for HR- and ER-inducing $R$ genes. Overall, the fitness of $SHR(+)$ plants can exceed that of $SHR(-)$ plants when propagation occurs in a highly local manner and thus can be evolutionarily stable, at least under specific circumstances, including limited but constant entry of viruses from outside the plant population and local virus transmission.

Recent studies have revealed that virus infection sometimes has beneficial effect on host plants at least in certain virus–host combinations and in certain circumstances, such as under abiotic stresses[39]. We simulated such situations by giving a lower mortality rate to infected $SHR(-)$ plants ($d_{ni} = 0.08$) than non-infected plants ($d = 0.1$). The simulation resulted in rapid decrease in occupancy of $SHR(+)$ plants [i.e., increase in occupancy of $SHR(-)$ plants] irrespective of plant propagation mode (defined by $l_p$) or initial occupancy of the genotypes (defined by $R_{SHR+}$) (Supplementary Fig. 12).

## Discussion

**MOI decrease as an individual-level resistance mechanism.** Previous studies have suggested PCD-independent containment of viruses in $R$-gene-mediated resistance[5–7,40–43]. Consistent with these findings, we detected a decrease in viral MOI upon induction of HR or SHR after the first and second cell-to-cell infection, when PCD was still not induced. During induced HR, viral MOI decreased by 28.7% for the first and second cell-to-cell movement combined. In theory, viral MOI in HR decreases until it reaches zero, when the virus stops spreading to adjacent cells. Due to technical limitations, we can currently provide reliable MOI estimates for only the first and second cell-to-cell movements. Future studies may solve this problem and detect a consistent decrease in MOI with additional cell-to-cell infection. Salicylic acid-depleted plants allowed further expansion of infected regions in HR-induced lesions[7]; this finding implies that salicylic acid is at least partially involved in the plant-mediated decrease of viral MOI. The observed decrease in MOI upon SHR induction was moderate (12.7%), demonstrating that the level of resistance induction is lower in SHR than in HR. As shown in our results, this relatively small difference (i.e., 28.7% vs. 12.7% MOI reduction) can change the fate of infected plants, by determining whether they succeed in containing the virus or fail to stop the expansion of infected regions, resulting in systemic death caused by PCD. This type of systemic death in SHR can also function as a population-level resistance mechanism (discussed later.)

Regarding the molecular mechanisms that may explain the observed decreases in MOI, we previously proposed that plant viral MOI could be determined by a simple formula:

$$\mathrm{MOI} = E \cdot R \cdot p/d \qquad (1)$$

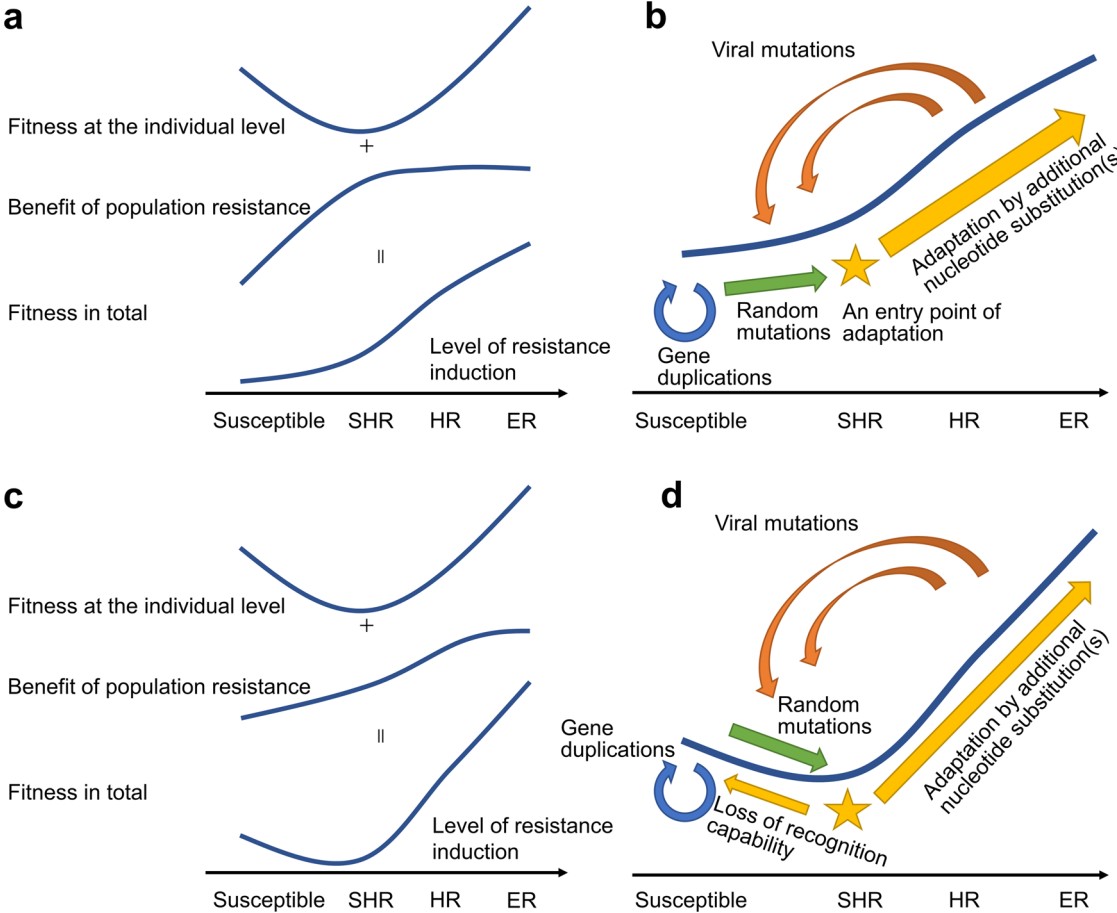

**Fig. 7 Proposed evolutionary trajectory of antiviral _R_ genes.** Schematic representations for fitness curves and evolutionary trajectories when the benefit of suicidal population resistance exceeds the cost of individual death (**a**, **b**) and does not (**c**, **d**). Fitness at the individual and population levels for full susceptibility and the induction of SHR, HR, and extreme resistance (ER) (**a**, **c**). Proposed trajectory of antiviral _R_ gene evolution (**b**, **d**). Note that adaptation from SHR to full susceptibility will also occur in the latter situations.

where _E_ is the number of viral genomes introduced to a host cell, _R_ is the number of sites for replication complex (RC) formation in a cell, _p_ is the probability that a viral genomic RNA forms an RC at one site in a unit of time, and _d_ is the degradation rate of viral genomic RNA in a unit of time[37]. Any MOI decrease upon host resistance induction can be explained by changes in these parameters. For example, _p_ may decrease through less efficient translation of viral proteins; _d_ may increase through enhanced activity of the RNA-silencing pathway, e.g., _RDR1_ induction by salicylic acid[44]; and _E_ may decrease through inefficient accumulation of viral genomes in the previous cell and/or by callose deposition at plasmodesmata[45]. Estimations of MOI in knockdown or knockout plants for related genes would help to quantify the contribution of each factor to virus containment. This reductionist understanding of MOI decreases also allows us to discuss ER in the same context. In ER, viral accumulation stops below the threshold for PCD induction in an initially infected cell, which can be explained by a rapid decrease in _p_ and/or rapid increase of _d_ in the infected cell. Future studies may address common molecular mechanisms in virus containment in HR and ER.

**_R_-gene-mediated MOI decrease targets the Achilles heel of plant viruses.** Previously, we proposed that a small MOI during cell-to-cell movement has critical importance for plant viruses because it results in stochastic separation of adaptive and defective viral variants co-existing in intracellular populations, leading to selection of the most adaptive variants[36,46,47]. This theory is supported by many previous[36,37,48,49] and current observations that have indicated small MOIs during cell-to-cell infections in the absence of _R_ genes. In the current study, we observed a decrease in viral MOI in the presence of an _R_ gene; we suggest that this decrease can explain virus containment in infected leaves. Notably, such a resistance mechanism would never be functional if the viral MOI in susceptible plants, which lack an _R_ gene, was originally high. For example, it would be much more difficult for a plant to employ defense mechanisms to decrease the viral MOI from 100 to 0 than to decrease the MOI from 5 to 0. Therefore, we conclude that a small MOI is a predestined weak point, i.e., the Achilles heel of plant viruses, which is targeted by _R_-gene-mediated resistance in plants. In this context, small MOIs are a good target for the development of artificial antiviral strategies in plants as well.

**The development of an antiviral suicidal population resistance in land plants.** Our simulations imply that PCD-mediated systemic death in SHR can function as a suicidal virus elimination mechanism that saves closely related kin from viral infection, contributing to a mechanism for population-level resistance. As our simulation model does not employ observation-based quantitative parameters such as distance between plants or reach of insect vector flight, further quantitative simulations involving such parameters are required to establish if SHR could have positive fitness consequences for neighboring plants in nature;

however, our simulations showed that the suicidal population resistance prevailed only under strict local host propagation, and is therefore valid only for organisms with limited locomotion and dispersal, such as land plants. This suicidal population resistance may not be very suitable as a resistance mechanism against CMV, because the virus has an exceptionally broad host range, and thus the benefit of suicidal death may be shared by other species; however, because most plant viruses are thought to have much narrower host ranges[50], suicidal population resistance may function against such viruses in general. Further studies are required to discuss the impact of host range, because currently known host range for each virus may differ from an actual one[51]. Of note, a marked increase in the copy number of *R* genes occurred during land plant evolution[52]. This probably reflects the increase in the number of organisms in contact with land plants; suicidal population resistance might have allowed land plants to broaden the resistance spectrum against viruses by *R* gene multiplication. Although PCD-dependent cell death is observed in HR, several studies have shown that cell death in HR is neither required nor sufficient for the prevention of systemic viral infection[5,7] and that the pathways that induce PCD or contain viral infection are independent[43,53,54]. Considering that a suicidal population resistance incorporating SHR requires cell death to eliminate viral infection source, PCD-mediated cell death may play a role in population-level resistance rather than in individual-level resistance, in both SHR and HR against viruses. Local necrotic lesions formed in HR can decrease opportunities for vector transmission of the virus to other plants. This idea is compatible with the notions that PCD is activated only after a certain amount of viral protein has accumulated in an infected cell, and that ER, a stronger resistance that significantly reduces viral accumulation at the cellular level, is not associated with necrosis.

A pioneering study by Fukuyo et al. theorized that a suicidal population resistance could be beneficial for single-cell organisms based on simulations and corresponding experiments in a bacterium–bacteriophage system, and demonstrated that local propagation was required for such a resistance mechanism[55]. The conclusions obtained in our plant–virus model are consistent with their conclusions; we add that such a suicidal population resistance can be suitable for explaining *R*-gene-mediated resistance in land plants, whereby suicide under local propagation can help the protected plant population to gain time to obtain *R*-gene variations that confer stronger resistance to the virus, despite lower individual plant fitness for suicide than for full susceptibility.

Recent studies have shown that viral infection sometimes confers host tolerance to abiotic stresses in certain virus–plant combinations[39]. Our simulation suggested that full susceptibility is favored in such friendly interaction (Supplementary Fig. 12). It is arguable how general such situations are; the accumulation of many *R* genes in plant genomes in turn emphasizes that hostile interactions between plants and microbes are not rare.

Field studies on lupin crops and a systemic-necrosis-inducing strain of bean yellow mosaic virus showed that systemic death inhibited polycyclic plant-to-plant viral spread, which occurred for a non-necrosis-inducing strain of the virus used as a control; based on the results, the authors proposed the use of the systemic necrosis as a population-level field resistance[56,57]. Although the current study focuses on the evolution of suicidal population-level resistance in nature, the field resistance concept has an overlap with our discussion in that neighboring plants can be protected by systemic necrosis. Because of this overlap, our simulation model has a compatibility with crop field situations: for example, expected failure of population resistance at an enhanced influx of the virus (Fig. 6b) may also occur in crop fields; the different

levels of local propagation in our model can be related to the size of crop fields. Thus, our simulation model can be modified to discuss the practicability of field resistance.

**A theory for the trajectory of antiviral *R*-gene evolution in nature.** Based on these findings, we summarize possible trajectories of antiviral *R*-gene evolution in nature (Fig. 7). Plant populations that are susceptible to a certain pathogenic virus may obtain a new *R* gene, via gene duplication and random mutations in existing *R* genes, that can recognize the virus. The level of resistance induction may be insufficient to induce HR or ER, instead inducing SHR. An SHR-inducing trait causes lower fitness than full susceptibility at the individual level; however, the same trait can increase plant fitness at the population level, only because the local propagation of land plants promotes suicidal resistance. The benefit of this population resistance may or may not be big enough to exceed the cost of individual death, depending on the mode of plant propagation, viral host range, and the mode of viral transmission (Fig. 7a, c). In both cases, the benefit of the population resistance promotes SHR as entry points for *R*-gene adaptation toward HR induction, though selection toward full susceptibility may also occur in the latter case (Fig. 7d). Subsequent *R* gene adaptation toward HR could occur, for example, through additional nucleotide substitutions in the *R* gene that increase the induction level of resistance. Such genetic conversions can also change an HR-inducing gene to an ER-inducing gene. Considering that ER-inducing *A. thaliana* (*R++*) plants did not allow accumulation of the HR-breaking CMV CP-T45M variant, an ER-breaking virus variant may not occur as frequently as an HR-breaking variant; in this context, ER induction is more adaptive than HR induction, and therefore will be further selected in nature. Viral mutations in the recognized protein may decrease the level of induced resistance, changing HR- or ER-inducing plants to SHR-inducing or susceptible plants, and beginning another round in the *R*-gene adaptation cycle. These insights into possible trajectories of antiviral *R* gene evolution will benefit future studies on evolution histories of plant–pathogen interactions, as well as studies on agricultural application of *R* genes.

## Methods

**Plants.** The chimeric *RPRPCY* transgene for *N. benthamiana* was previously described[28]. *A. thaliana* plants with different copy numbers of *RCY1* transgenes were described previously[9,29]. *N. benthamiana* plants were grown for 4 weeks in a growth chamber at a stable temperature of 25 °C under a 14-h photoperiod. *A. thaliana* plants were grown for 6 weeks in a growth chamber at a stable temperature of 23 °C under a 10-h photoperiod.

**Viral cDNA constructs, in vitro RNA transcription, and inoculation.** cDNA constructs for WT CMV(Y) segments, pCY1, pCY2, and pCY3, were described previously[58]. cDNA constructs for RNA2 fluorescent protein variants were prepared by inserting YFP or CFP cDNA[36] into pC2-A1[59]. The mNLS sequence (PEPPKKARKVEL; underlined position has an Ala residue instead of a WT Lys residue for moderate nuclear localization efficiency) was designed based on that of Hodel et al.[60]. The resultant cDNA constructs were named pCY2-mNLSYFP and pCY2-mNLSCFP. cDNA constructs for the RNA3 variants were prepared by site-directed mutagenesis. After linearization with appropriate restriction enzymes, each cDNA was used as a template for in vitro transcription with AmpliCap-Max T7 High Yield Message Maker Kit (Cellscript, US). For inoculation of *N. benthamiana* plants, equal amounts of transcripts were mixed and inoculated to plant leaves using carborundum as an abrasive. For inoculation of *A. thaliana* plants, we first inoculated in vitro transcripts to WT *N. benthamiana* plants and collected infected leaves after 4 days. Infected leaves were ground in 100 mM potassium phosphate buffer (pH 8.0) and the sap was used to inoculate *A. thaliana* plants.

**Detection of CMV infection.** Systemic infection of *N. benthamiana* plants with YFP-tagged virus was visualized by detecting YFP fluorescence using blue excitation light and a longpass filter (>515 nm). Systemic infection of *A. thaliana* plants was analyzed by immunological detection of viral CP with press blotting and western blotting as described previously[3]. Briefly, aerial parts of the plants were

harvested for press blotting between two filter papers (3MM CHR, Cytiva, US.) The blotted filter papers were shaken in 2% Triton X-100 to remove green color, followed by blocking in PBST [10 mM sodium phosphate buffer (pH 7.2), 0.9% NaCl, and 0.1% Tween-20] supplemented with 3% skim milk for 30 min. The filter papers were further incubated for 1 h with a primary antibody [an inhouse raised rabbit antibody against CMV(Y)] added at 1:10,000 dilution. After washing with PBST-skim milk for three times, the filter papers were incubated with a secondary antibody [alkaline phosphatase-conjugated anti-rabbit IgG (Fc), Promega, US; diluted at 1:10,000] for 1 h, followed by washing three times with PBST-skim milk and three times with AP9.5 buffer [10 mM Tris-HCl (pH 9.5), 100 mM NaCl, and 50 mM MgCl$_2$]. Detection was done with NBT-BCIP solution (0.03% nitro blue tetrazolium and 0.015% 5-bromo-4-chloro-3-indolyl-phosphate dissolved in AP9.5 buffer.) For western blot analysis, three leaf discs (6 mm diameter) were homogenized in 50 μl sample buffer [50 mM Tris-HCl (pH 6.8), 2% sodium dodecyl sulfate, 6% 2-mercaptoethanol (v/v), 10% glycerol (v/v), and 0.025% bromo phenol blue] and boiled for 3 min immediately. Two microliter each of the samples was run in 10% SDS-PAGE gel. The proteins were blotted to nitrocellulose membranes by a semidry method with a blotting buffer [0.3% Tris, 1.4% glycine, and 20% methanol (v/v)]. The membranes were blocked by a homogenate of healthy *N. benthamiana* leaves in TTBS [20 mM Tris-HCl (pH 7.5), 150 mM NaCl, and 0.05% Tween-20] for 30 min, followed by 1-h incubation with a primary antibody against CMV(Y) diluted at 1:10,000. After washing three times with TTBS, the membranes were incubated with a secondary antibody [alkaline phosphatase-conjugated anti-rabbit IgG (Fc), Promega, US, diluted at 1:10,000] for 1 h. The membranes were washed twice with TTBS and once with AP9.5 buffer, prior to the detection with NBT-BCIP solution. Total proteins were detected by staining the gels in a CBB staining solution [0.1% CBB-R250, 5% methanol (v/v), and 7% acetic acid (v/v)] for 3 h and destaining them in a CBB destaining solution [5% methanol (v/v) and 7% acetic acid (v/v)] overnight.

**DAB staining**. Inoculated *N. benthamiana* leaves and uninoculated upper leaves were harvested at 5 days after inoculation and vacuum infiltrated with 1% DAB solution in distilled water. After 2-h incubation at room temperature in dark, the leaves were moved to 100% ethanol and boiled for 10 min with a single-time substitution of the ethanol.

**Genomic DNA fragmentation**. Genomic DNA was extracted from inoculated leaves and uninoculated upper leaves of *N. benthamiana* at 5 days after inoculation (before necrosis become visible) by grinding in DNA extraction buffer (200 mM Tris-HCl pH 7.6, 250 mM NaCl, 25 mM EDTA, and 0.5% SDS) and purification by standard phenol/chloroform extraction. After RNaseA treatment and phenol/chloroform extraction, the concentration of extracted DNA was adjusted using a Qubit fluorometer (Invitrogen, US.) Approximately 150 ng DNAs were subjected to agarose gel electrophoresis. We used ImageJ version 1.52p for quantification of signal intensities.

**MOI estimation**. For MOI estimation, WT or *R* gene-transformant *N. benthamiana* plants were inoculated with a mixture of RNA transcripts consisting of WT RNA1, RNA2-mNLSYFP, RNA2-mNLSCFP, and different RNA3s (i.e., WT or a single-amino acid variant). At 14–16 h after inoculation, YFP and CFP fluorescence was observed under a fluorescence microscope (AxioImager.M1, Zeiss, Germany) equipped with appropriate filter sets and a charge-coupled device camera (AxioCam MRm, Zeiss, Germany). Using Adobe Photoshop Elements version 15.0, YFP and CFP fluorescence images were converted to green and magenta, respectively, and then merged to determine whether each site/cell contained both YFP and CFP fluorescence. Data analysis was performed as described previously[36,37], with the modified assumption of equal MOI for the first and second cell-to-cell movements. Briefly, the number of co-infected and singly infected cells was obtained from co-infected sites with 10–30 cells (i.e., those with Cell-0 + Cell-1 + Cell-2 cells) and most likely MOI that reproduces the observed frequencies of co-infected and singly infected cells was estimated by maximum-likelihood method using the R software[61]. The R script used for the MOI estimation is available as Supplementary Text 1, as well as at Github (https://github.com/ShuheiMiyashita/Suicidal_Resistance2021) and at Zenodo[62]. Note that we did not use MOI estimates for Cell-0 infections to discuss anything, because Cell-0 can be influenced by inocula concentration and our mechanical inoculation efficiency, and thus have less biological meaning. In other words, Cell-1 and Cell-2 MOI estimates are not affected by such artificial or technical fluctuations because Cell-1 and Cell-2 infections occur by natural cell-to-cell movement from Cell-0 and Cell-1, respectively. The script simultaneously estimates Cell-0, Cell-1, and Cell-2 MOIs, because Cell-0 MOI estimates allow us to exclude the effect of the above artificial and technical fluctuations in MOI estimations for Cell-1 and Cell-2. Exclusion of Cell-0 MOI estimates from discussion also helps avoid touching the possible difference in initial infection routes between natural aphid-borne infections and mechanical inoculations employed in this study.

**Simulation**. The model used for the simulation is explained in the Results section. Simulations were run using the R software; the R script used for the simulation is available in Supplemental Information (Supplementary Texts 2–5) as well as at

Github (https://github.com/ShuheiMiyashita/Suicidal_Resistance2021) and at Zenodo[62].

**Statistics and reproducibility**. For MOI estimation, we know a relation that analyzing four times more cells will give half standard deviation for MOI. Thus, we could calculate the number of cells need to be analyzed to have less than 10% of standard deviation to the MOI estimates. As the number of infected sites per inoculated leaf fluctuates to some extent, we inoculated the plants on several independent days until we have enough number of cells. We estimated MOI and its standard deviation for each day, and confirmed that estimated MOI of a certain day appears within the standard deviation of the other day(s); this confirms that the MOI estimates of independent days are not significantly different. After that we combined those data of independent experiments and obtained an MOI estimate that appear in the Results section.

**Reporting summary**. Further information on research design is available in the Nature Research Reporting Summary linked to this article.

## Data availability
The authors declare that all the data supporting the findings in the current study are available within the article and its Supplementary information files. Uncropped and unedited bright-field and fluorescence images and blots and gel images can be found as Supplementary Figs 13–25 in the end of Supplementary information PDF file. Source data for Supplementary Fig. 3d are provided as Supplementary_data.xlsx. The cDNA sequences of pCY2-mNLSYFP and pCY2-mNLSCFP are deposited in DDBJ/EMBL/GenBank under the accession IDs of LC602828 and LC602829, respectively. All other relevant data relating to this manuscript are available from the corresponding author upon reasonable request.

## Material availability
Commercially unavailable materials can be obtained from the corresponding author upon reasonable request, through a material transfer agreement with Tohoku University.

## Code availability
All the R scripts used for MOI estimation and simulation analysis are provided in Supplementary information PDF file (as Supplementary Texts 1–5) as well as at Github (https://github.com/ShuheiMiyashita/Suicidal_Resistance2021) and at Zenodo (https://doi.org/10.5281/zenodo.5105622).

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

## Acknowledgements

We thank Drs Chikara Masuta, Masashi Suzuki, and Minoru Takeshita for providing the pCY1, pCY2, pCY3, and pC2-A1 constructs. The super-computing resource was provided by Human Genome Center, the Institute of Medical Science, the University of Tokyo. This work was partially supported by Japan Society for the Promotion of Science

(JSPS) KAKENHI grants (16H06185, 17K19257, 19H02953, and 21K05591); Ministry of Education, Culture, Sports, Science and Technology (MEXT) "Scientific Research on Innovative Areas" grants (16H06429, 16K21723, and 16H06435); and a JSPS Core-to-Core Program (Advanced Research Networks) entitled "Establishment of an International Agricultural Immunology Research Core for a Quantum Improvement in Food Safety." The funders have no role in study design, data collection and analysis, decision to publish, or preparation of manuscripts.

## Author contributions

S. M. designed the study. D. A. A., S. v. B., M. S., and S. M. carried out the experiments. D. A. A., S. A., H. T., and S. M. analyzed the data. S. M. developed the simulation model and interpreted the results. D. A. A., S. v. B., and S. M. wrote the paper.

## Competing interests

The authors declare no competing interests.
