## [Peer review file · Communications Biology]

Reviewers' comments:

Reviewer #1 (Remarks to the Author):

The authors of this manuscript describe a system in which one of four host responses occurs (a compatible systemic infection, an HR, an ER or an SHR), depending on the presence and copy number of a resistance gene, and on differences in the viral CP sequence. They examine the multiplicity of infection (MOI) of the virus variants involved in the various host (trans) genetic backgrounds and deduce that there is a relationship between the MOI and the nature of the host response – HR or SHR – with a much greater depression of the MOI in plants showing an HR than in those showing an SHR. They then do simulation studies of infections to provide evidence for their hypothesis that SHR is a form of population control of infection, where the needs of the many outweigh the needs of the few. Regarding this study, there are three issues of concern for this reviewer:

(1). In establishing the MOI, it is not clear how the authors established a minimum concentration of inoculum to use for this study, or how the concentration of inoculum used would affect their results. In addition, since the virus is normally transmitted by aphids, how does the inoculum applied by rubbing a leaf relate to a natural infection, with a very different dose and potential inoculation route, with numbers sites of inoculation by numerous vectors? These questions relate to the relevancy of the inoculation study to the real-world experience.

(2). Starting on In 22 and then throughout the manuscript there is an assumption that SHR is plant cell death (PCD, aka apoptosis). PCD, by definition is a programmed process, involving nuclear blebbing and DNA fragmentation, as well as other responses. In contrast, cellular or tissue necrosis can happen by various responses, including oxidation reactions leading to damage to membranes, which can lead to vascular cell damage causing distal cell death due to loss of water movement. Have the authors shown that cell death in either their system or any they refer to has been shown to be due PCD rather than just necrosis?

(3) There are many papers (mostly by breeders) with descriptions of resistance responses to infection by viruses in crops (and sometimes the genetic inheritance) that involve resistance of the SHR type, although these papers just refer to such as systemic necrosis, or more often as 'field resistance'. Thus, this concept is well known, if not well referenced in this work. More references to work further afield to the narrow spectrum cited here are needed.

Lesser issues:

1. In 2. Delete 'the' before 'kin'.

2. In 26-28. This is unclear here as to the respectivity, given that three parameters are involved: MOI number, resistance response, and virus variation. Rephrase.

3. In 96. CMV was already abbreviated on In 47.

4. In 98. First use of 'CP' (undefined). It is not defined until In 592 (in the Fig. 1 legend).

5. In 124. Citing unpublished data requires all authors involved to be named with their family name and first initial, and not just et al., in case those authors include others not listed as authors of the current manuscript.

6. In 337. As for In 124.

7. In 413. Here, Alanine and lysine are written out (one with a capital initial letter), while earlier, the three letter abbreviations for amino acids were used. The choice of which to use is arbitrary (unless a journal has a specific policy), but should be consistent throughout a study.

Reviewer #2 (Remarks to the Author):

The authors present very interesting biological experiments, thought experiments and model simulations to back up the latter to make the case that systemic hypersensitive responses (SHRs) can have a defensive function at the population level, despite having negative consequences for the

individual plant that displays this phenotype.

I have no qualms about the biological experiments the authors performed. They have carried out similarly elegant experiments to estimate the multiplicity of infection (MOI) in two other plant-virus interactions over the years: Japanese soil-borne wheat mosaic virus in *Chenopodium quinoa* (Miyashita and Kishino, *Journal of Virology*, 2010), and tomato mosaic virus in *Nicotiana tabacum* (Miyashita et al., *PLoS Biology*, 2015). And indeed, the interactions of cucumber mosaic virus (CMV) with *Arabidopsis thaliana* and *Nicotiana benthamiana* are well-established genetic model systems for studying general properties of viral infections.

Although I am not an expert in simulation modeling, to the extent that I can evaluate the model's set up, everything looks to have been done appropriately to me from a technical point of view. That said, there is one parameter that appears to be missing in my opinion – please correct me if it is there and I have misinterpreted some of the lines of code – and that is a parameter to account for (within-host) virus evolution.

If such a parameter is indeed not included, then I feel that this should be to reflect the biological differences between virus and host plant in the accessibility of mutations that may provide switching between different phenotypic outcomes of interactions. The virus has better potential access to such mutations as indeed reflected by the authors' discovery of a viral mutation in the form of T45M that allowed the virus to escape from HR. And perhaps such a virus evolution parameter should be split in two separate parameters, one that covers a rate with which virus mutations bring the phenotypic outcome towards SHR and another one that covers a rate with which virus mutations take the phenotypic outcome away from SHR. I wonder how sensitive the outcomes of the simulations would be to the inclusion of virus evolution. I realize that the model is more in the category of epidemiological models rather than a model over evolutionary time-scales, but given the biologically realistic opportunity for viruses to evolve within one life-history cycle of the host inclusion of a virus evolution parameter might be necessary to model the spread of infections in a meaningful way.

Apart from this, I do see the added value of this modeling effort and I am glad the authors have decided to take this route. One of the reasons that I am enthusiastic is that I have seen the potential for SHR occurring in natural settings with my own eyes and I have been curious about potential implications. I have observed SHR myself in the interaction between CMV-Fny, another naturally occurring strain of CMV that is slightly different from CMV-Y (but in the same CMV subgroup), and the natural *Arabidopsis* accession Co-1 (from Coimbra, Portugal). Co-1 is closely related to accession C24, which carries the RCY1 allele that triggers a HR in interaction with CMV-Y and which the authors are studying in detail.

Having made my comments about the biological experiments and the model simulations, I will now move on to the authors' biological interpretations of their results. There are three reservations that I have with the biological interpretations, and in my view edits to the manuscript are necessary to keep open the possibility at the very least that the results the authors observe are not indicative of kin selection at play, and the biological interpretations may need to be changed altogether.

My reservations are: 1) how the authors describe the occurrence of SHR as a 'deliberate' strategy, 2) how the authors invoke kin selection, and 3) how the authors seem to assume that susceptibility to virus infections can only have negative fitness consequences and be selected against. I will describe these reservations in more detail:

1) In the authors' experiments the 'ancestral' state of the interaction between plant and virus is full resistance of the plant, with the plant presumably showing 'high' fitness (although the authors did not measure this). In this biological scenario in which the plant 'loses' a beneficial phenotype (virus resistance) it requires an enormous jump in logic to see losing this beneficial phenotype as a strategy by the plant, but I will get back to this in my second and third points. Also, the word "strategy" makes

it sound like natural selection has a purpose and works deliberately towards certain goals like an engineer, which is of course not the case. Mismatches between host and virus and even between different genotypes of the same host plant species occasionally lead to systemic necrosis (or 'hybrid' necrosis in the latter case; e.g., Bomblies et al., PLoS Biology, 2007), and applying Occam's razor a more parsimonious interpretation might be that these are evolutionarily unstable states at maladaptive locations in a fitness landscape that will be resolved towards more stable states at adaptive peaks as new mutations occur or as ecological conditions and thus locations of fitness peaks change over time. There may not be a need to invoke the less parsimonious scenario of kin selection, particularly in the case of CMV, and I will go into this now.

2) My interpretation of SHR, which I think is more parsimonious and more generally applicable, is that SHR, when it occurs in certain plant-genotype-by-virus-genotype combinations, is a manifestation of phenotypic plasticity that has the effect of limiting the strength of natural selection on resistance in the host population; i.e., how strongly the host population is pushed to move towards either full immunity or to full susceptibility (more on susceptibility in point 3 below).

The way SHR might constrain selection is by removing potential sources of virus inoculum from a population or community of plants, thereby limiting the strength of selection that the virus as an agent of selection poses on any susceptible plants in the area. The strength of selection on the virus to mutate further to bring the system towards susceptibility is still high because the plant undergoing SHR is still a dead-end host plant for the virus.

The occurrence of SHR might potentially be interpreted as a scenario of kin selection when the host plants are reproducing clonally, through selfing, or through severe inbreeding, when all plants of a population in a stand are genetically identical or nearly so, and when the virus is a specialist with a very narrow host range. This places very severe restrictions on when an interpretation of kin selection might be warranted. And even then, I would favor a more generally valid biological interpretation of viewing SHR as a plastic phenotypic outcome that constrains natural selection on immunity.

In the case of CMV in particular the above-mentioned assumptions of when kin selection might potentially be invoked will be extremely rare in natural settings. The virus is a generalist virus that infects more than 1,200 plant species. Chances are slim that a population of one of these host plant species would grow in monoculture in natural settings. However, the occurrence of SHR in one host population would still remove potential sources of virus inoculum from the community of susceptible plants, although community connections also depend on the host ranges of virus-vectoring aphids. SHR would thus influence patterns of selection on resistance in multiple plant populations in this community regardless of whether they are kin of or more distantly related to the infected plant. The occurrence of SHR in one plant species could thereby even constrain selection on resistance to CMV in populations of other plant species.

The descriptions of the model simulations should make more mention of any limitations on the ecological settings to which the simulations may apply.

In my opinion, the "inefficient induction of antiviral R-gene-mediated resistance" as in the title, should not be viewed "as a suicide strategy to save the kin from the virus", but could more appropriately be viewed as phenotypic plasticity that constrains natural selection for resistance or susceptibility.

3) This last point brings me to another aspect and that is how the authors seem to assume that susceptibility to virus infections can only be selected against. Depending on the ecological settings selection may act towards favoring susceptibility. More and more examples of infections having positive fitness consequences depending on the identity of plant and virus and the ecological settings are being published (e.g., Groen et al., PLoS Pathogens, 2016; Xu et al., New Phytologist, 2008). And as mentioned above, also here, lowering the number of infection sources would change patterns of selection for susceptibility just as much as they would change patterns of selection for full immunity.

I agree with the authors that SHR can form an entry point for R-gene adaptation as they mention in the Discussion, but I do not agree that this is necessarily a "stable" entry point. Depending on the ecological settings and the phenotypic and fitness consequences of a fully susceptible interaction, the population may just as well revert from SHR towards full virus susceptibility.

Overall, I think the authors have put together an intellectually stimulating manuscript based on a thorough effort of biological experiments and modeling simulations. The manuscript was written very well, which made the authors' ideas not difficult to understand.

Reviewer #3 (Remarks to the Author):

The authors Abebe et al present interesting idea on the evolution of plant – virus interaction. They support is with studying infectivity of viral mutants in diverse plant genetic backgrounds and with population level model simulations. There are some minor points to correct and improve.

- l61: there seem to be different molecular mechanisms leading to ER, see also <https://onlinelibrary.wiley.com/doi/full/10.1111/pbi.13230>. Also the further explanations (till l66 is maybe not appropriate for the introduction as it describes your opinion. Move it to discussion
- l136-138: please specify exactly which combinations give HR and which ER, this is also not visible in figure 3 , please provide better images showing necrosis
- l144: Plant RNA viruses accumulate 106–107 copies in each infected cell (e.g., Takebe and Ohtsuki, 1969). – this is very old citation and covers only one pathosystem. At least our data show more than 10-fold differences in different viral pathosystems. Please recheck the generality of the statement
- definition of MOI is well described in results section, thus I propose to modify it in l26 to The average number of viral genomes that establish infection in next cell
- the manuscript Qu et al is not yet published, thus we cannot evaluate it's relevance for this paper
- L352-353: From this sentence one understands that it is PCD that is killing the virus. Which is not true, it is a side effect probably aiming at other pathogens. My suggestion is to change it to ' Our simulations imply that SHR response can function as a suicide strategy to save closely related kin from viral infection, contributing to a mechanism for population-level resistance.'
- l358-359: 'The invention of a suicide strategy may be a plausible explanation for this expansion in the number of R genes in land plants.' Please rephrase this part as there are other more direct explanations for the expansion in number of R-genes. For sure they are evolving to cope with multitude of organisms they are in contact with
- l364: Considering that a suicide strategy incorporating SHR requires cell death,... You have not proved that cell death is required, only that the plant needs to die. The plants die also without SHR, in 'normal' compatible interaction. Please rephrase
- authors should consider the rewriting of the abstract as the message of their achievements was for me not clear when reading the abstract before the manuscript

Point-to-point answers to Reviewers' comments:

Reviewer #1 (Remarks to the Author):

The authors of this manuscript describe a system in which one of four host responses occurs (a compatible systemic infection, an HR, an ER or an SHR), depending on the presence and copy number of a resistance gene, and on differences in the viral CP sequence. They examine the multiplicity of infection (MOI) of the virus variants involved in the various host (trans) genetic backgrounds and deduce that there is a relationship between the MOI and the nature of the host response – HR or SHR – with a much greater depression of the MOI in plants showing an HR than in those showing an SHR. They then do simulation studies of infections to provide evidence for their hypothesis that SHR is a form of population control of infection, where the needs of the many outweigh the needs of the few.

Regarding this study, there are three issues of concern for this reviewer:

(1). In establishing the MOI, it is not clear how the authors established a minimum concentration of inoculum to use for this study, or how the concentration of inoculum used would affect their results. In addition, since the virus is normally transmitted by aphids, how does the inoculum applied by rubbing a leaf relate to a natural infection, with a very different dose and potential inoculation route, with numerous sites of inoculation by numerous vectors? These questions relate to the relevancy of the inoculation study to the real-world experience.

We thank the reviewer for raising the important points. We have added more explanation for our MOI estimation procedure (L559-569/L539-549). In bacteriophage or animal cell culture experiments, it is true that viral MOI can be artificially controlled by changing the concentration of the inocula; however, for MOIs in plant virus cell-to-cell movements (infection of cell-1 and cell-2 in Fig. 4d), we cannot control the virus concentration because the viral genomes come from neighboring infected cells (cell-0 and cell-1, respectively); thus, the MOIs here reflect natural infections. On the other hand, MOI for the initial cell (cell-0) can be changed by inoculum concentration and our inoculation efficiency, as in bacteriophage experiments; because of this reason, we do not use the cell-0 MOI estimates to discuss anything (we modified the manuscript to explicitly touch it: L559-561/L539-541). The reason why we still estimate cell-0 MOI in our script (Text S1) is that we can know and remove the effect of the above technical fluctuation from cell-1 and cell-2 MOI estimations (This is implemented by making table0 and table0m in the script).

CMV is transmitted by aphids; however, as discussed above, our MOI estimates

for cell-1 and cell-2 infections are not influenced by infection route of cell-0. So, we consider the relevancy of our MOI estimates to real world is not affected.

(2). Starting on Ln 22 and then throughout the manuscript there is an assumption that SHR is plant cell death (PCD, aka apoptosis). PCD, by definition is a programmed process, involving nuclear blebbing and DNA fragmentation, as well as other responses. In contrast, cellular or tissue necrosis can happen by various responses, including oxidation reactions leading to damage to membranes, which can lead to vascular cell damage causing distal cell death due to loss of water movement. Have the authors shown that cell death in either their system or any they refer to has been shown to be due PCD rather than just necrosis?

We thank the reviewer for this careful discussion. Below I separate the points and discuss:

1) *R*-gene-dependent systemic death

Because CP-T45M mutant does not cause necrosis in *N. benthamiana* or *A. thaliana* plants without *R* gene (Fig. 2e and Fig. S5 added upon reviewer 3's comment), we consider that the death of upper leaves occurred in an *R*-gene-dependent manner. To support *R*-gene-dependent systemic death, we added Fig. S2, which shows that CP-N31T&T45M double mutant does not cause systemic death when infecting *R* gene (+) plants. We added some sentences to explain this part (L148-151/L136-139).

2) Evidence for programmed-cell death

Unfortunately, there is no golden standard to evidence programmed-cell death (PCD) induction in plants, but we added DAB staining and genomic DNA electrophoresis (Fig. S3) to demonstrate the similarity of systemic death caused by CP-T45M mutant and an HR caused by wild-type virus, which support the occurrence of PCD in the former. We added the descriptions at L151-157/L139-143. Methods for the experiments were also added (L530-544/L510-524).

3) Timing of infection and death

We agree that cell death of vascular system can cause the death of distal cells; this is sometimes observed in our experiments as well. However, as we have demonstrated in Fig. S1 (newly added), we usually detect infection of upper leaves by YFP fluorescence first and then observe death of the infected region. We added the description to our manuscript (L139-141/L126-128).

(3) There are many papers (mostly by breeders) with descriptions of resistance responses to infection by viruses in crops (and sometimes the genetic inheritance) that

involve resistance of the SHR type, although these papers just refer to such as systemic necrosis, or more often as 'field resistance'. Thus, this concept is well known, if not well referenced in this work. More references to work further afield to the narrow spectrum cited here are needed.

We didn't touch such systemic necrosis because it is often unclear if it is *R*-gene dependent or not. However, we now agree that referencing such studies will broaden the scope of our study. We have added some examples in introduction (L75-99/L69-86,) and added a paragraph in Discussion to touch the conceptual overlap with "field resistance," together with some insights into such agricultural studies that could be obtained from our simulation results (L456-468/L437-449.)

Lesser issues:

1. In 2. Delete 'the' before 'kin'. We have followed this suggestion (L3/L3).
2. In 26-28. This is unclear here as to the respectivity, given that three parameters are involved: MOI number, resistance response, and virus variation. Rephrase. We have rephrased it (L25-29/L25-28).
3. In 96. CMV was already abbreviated on In 47. We have corrected it (L116/L104).
4. In 98. First use of 'CP' (undefined). It is not defined until In 592 (in the Fig. 1 legend). We have corrected it. Now CP appears first in L70/L64, for a PVY study.
5. In 124. Citing unpublished data requires all authors involved to be named with their family name and first initial, and not just et al., in case those authors include others not listed as authors of the current manuscript. We have modified the description accordingly. The result belongs to one of the authors (Takahashi, H.) and thus we have cited only his name (L146-147/L134).
6. In 337. As for In 124. Fortunately, the paper is now published. We have added its citation (L392/L377).
7. In 413. Here, Alanine and lysine are written out (one with a capital initial letter), while earlier, the three letter abbreviations for amino acids were used. The choice of which to use is arbitrary (unless a journal has a specific policy), but should be consistent throughout a study. We are sorry for this kind of careless mistakes. We have selected three letter abbreviations now (L509-510/L489; L129-130/L133).

We again thank the reviewer for careful and kind reading of the manuscript. We think our manuscript has been improved a lot by her/his comments.

Reviewer #2 (Remarks to the Author):

The authors present very interesting biological experiments, thought experiments and model simulations to back up the latter to make the case that systemic hypersensitive responses (SHRs) can have a defensive function at the population level, despite having negative consequences for the individual plant that displays this phenotype.

I have no qualms about the biological experiments the authors performed. They have carried out similarly elegant experiments to estimate the multiplicity of infection (MOI) in two other plant-virus interactions over the years: Japanese soil-borne wheat mosaic virus in *Chenopodium quinoa* (Miyashita and Kishino, *Journal of Virology*, 2010), and tomato mosaic virus in *Nicotiana tabacum* (Miyashita et al., *PLoS Biology*, 2015). And indeed, the interactions of cucumber mosaic virus (CMV) with *Arabidopsis thaliana* and *Nicotiana benthamiana* are well-established genetic model systems for studying general properties of viral infections.

We thank the reviewer for knowing our previous studies. Exactly as the reviewer describes, we use our experiment system as a model to study general properties of virus-plant interactions.

Although I am not an expert in simulation modeling, to the extent that I can evaluate the model's set up, everything looks to have been done appropriately to me from a technical point of view. That said, there is one parameter that appears to be missing in my opinion – please correct me if it is there and I have misinterpreted some of the lines of code – and that is a parameter to account for (within-host) virus evolution.

If such a parameter is indeed not included, then I feel that this should be to reflect the biological differences between virus and host plant in the accessibility of mutations that may provide switching between different phenotypic outcomes of interactions. The virus has better potential access to such mutations as indeed reflected by the authors' discovery of a viral mutation in the form of T45M that allowed the virus to escape from HR. And perhaps such a virus evolution parameter should be split in two separate parameters, one that covers a rate with which virus mutations bring the phenotypic outcome towards SHR and another one that covers a rate with which virus mutations take the phenotypic outcome away from SHR. I wonder how sensitive the outcomes of the simulations would be to the inclusion of virus evolution. I realize that the model is more in the category of epidemiological models rather than a model over evolutionary

time-scales, but given the biologically realistic opportunity for viruses to evolve within one life-history cycle of the host inclusion of a virus evolution parameter might be necessary to model the spread of infections in a meaningful way.

We totally agree that virus evolution is an attractive parameter, and of course we are also interested in it. However, in this study, we would like to keep focused on a question, “whether and on what condition can SHR benefit hosts?” The answer obtained from our simulation is “it’s possible when host propagations occur in a local manner.” We added more words to clarify this focus (L252-255/L239-242.)

In my opinion, modeling virus evolution is very difficult, because a single amino-acid substitution can drastically change the phenotype and such changes are difficult to expect and generalize. In a very simple simulation model where only one host and only one virus are assumed, virus will simply evolve to avoid *R*-gene-mediated recognition and decrease the pathogenicity on the other hand; however, once we start to make a realistic model, we need to model “unfortunate combinations” (like human and Ebolavirus and probably many known plant-virus combinations), which are difficult to generalize. So, we would like to leave this tough but attractive work to a future study that may include more field surveys.

Apart from this, I do see the added value of this modeling effort and I am glad the authors have decided to take this route. One of the reasons that I am enthusiastic is that I have seen the potential for SHR occurring in natural settings with my own eyes and I have been curious about potential implications. I have observed SHR myself in the interaction between CMV-Fny, another naturally occurring strain of CMV that is slightly different from CMV-Y (but in the same CMV subgroup), and the natural Arabidopsis accession Co-1 (from Coimbra, Portugal). Co-1 is closely related to accession C24, which carries the RCY1 allele that triggers a HR in interaction with CMV-Y and which the authors are studying in detail.

This sounds great to us as well. It would be interesting to focus on the difference in CMV-Y and CMV-Fny in our transformant *N. benthamiana* system. Because the CP of the two strains have only 6 aa differences, the different phenotype may reflect the difference in other characteristics (e.g. replication efficiency or cell-to-cell movement efficiency) that change MOI.

Having made my comments about the biological experiments and the model simulations, I will now move on to the authors’ biological interpretations of their results. There are

three reservations that I have with the biological interpretations, and in my view edits to the manuscript are necessary to keep open the possibility at the very least that the results the authors observe are not indicative of kin selection at play, and the biological interpretations may need to be changed altogether.

My reservations are: 1) how the authors describe the occurrence of SHR as a ‘deliberate’ strategy, 2) how the authors invoke kin selection, and 3) how the authors seem to assume that susceptibility to virus infections can only have negative fitness consequences and be selected against. I will describe these reservations in more detail:

We thank the reviewer for giving critical but constructive comments. We have read the comments carefully and improved the manuscript to make a fair discussion on our kin selection story and other stories.

Due to our insufficient description in the previous version of our manuscript, we may have misled the reviewer in some important points. For such points, we have modified our manuscript to clarify the points. Please see below for the details.

1) In the authors’ experiments the ‘ancestral’ state of the interaction between plant and virus is full resistance of the plant, with the plant presumably showing ‘high’ fitness (although the authors did not measure this). In this biological scenario in which the plant ‘loses’ a beneficial phenotype (virus resistance) it requires an enormous jump in logic to see losing this beneficial phenotype as a strategy by the plant, but I will get back to this in my second and third points.

It is true that in our experiments we started from HR phenotype and found a mutant virus that cause SHR; however, in our simulations, we test between SHR and full susceptibility. We added some description to avoid misleading the readers (L252-255/L239-242, already appeared above). Let us add that, in our simulation, we are neutral about the direction between SHR and susceptible; we just test which is more adaptive in different conditions.

Also, the word “strategy” makes it sound like natural selection has a purpose and works deliberately towards certain goals like an engineer, which is of course not the case.

We agree that natural selection never has a purpose – it’s just a phenomenon that more adaptive ones accumulate more. We changed the words “suicide strategy” to “suicidal population resistance” or similar words in the title and in other places accordingly (title and elsewhere.)

Mismatches between host and virus and even between different genotypes of the same

host plant species occasionally lead to systemic necrosis (or ‘hybrid’ necrosis in the latter case; e.g., Bomblies et al., PLoS Biology, 2007), and applying Occam’s razor a more parsimonious interpretation might be that these are evolutionarily unstable states at maladaptive locations in a fitness landscape that will be resolved towards more stable states at adaptive peaks as new mutations occur or as ecological conditions and thus locations of fitness peaks change over time. There may not be a need to invoke the less parsimonious scenario of kin selection, particularly in the case of CMV, and I will go into this now.

We also consider that SHR is not a fully adaptive situation (at least less adaptive than HR or ER); we noticed that our description that “SHR serve as a stable point” was not appropriate; we will discuss it later. As our simulation model suggests, SHR induction can be more adaptive than full susceptibility when propagation occurs in a local manner, while global propagation prefer full susceptibility. We consider that the fact that R genes accumulated in land plant genomes requires the occurrence of the former situation at least in some circumstances, and otherwise new R genes are hardly selected, especially because SHR induction can be a frequent phenotype, as observed in our experiments. We rewrote the discussion to include both situations (i.e., SHR induction is more and less adaptive than being fully susceptible,) and we added the latter case to Fig. 7 (L471-484/L452-464.)

2) My interpretation of SHR, which I think is more parsimonious and more generally applicable, is that SHR, when it occurs in certain plant-genotype-by-virus-genotype combinations, is a manifestation of phenotypic plasticity that has the effect of limiting the strength of natural selection on resistance in the host population; i.e., how strongly the host population is pushed to move towards either full immunity or to full susceptibility (more on susceptibility in point 3 below).

We agree that SHR can occur in certain plant-genotype-by-virus-genotype combinations; the reviewer and we have no dissidence on this point.

The way SHR might constrain selection is by removing potential sources of virus inoculum from a population or community of plants, thereby limiting the strength of selection that the virus as an agent of selection poses on any susceptible plants in the area. The strength of selection on the virus to mutate further to bring the system towards susceptibility is still high because the plant undergoing SHR is still a dead-end host plant for the virus.

This situation (that R-gene-mediated virus removal stop the selection) was also

observed in our simulations, when we assumed one-time introduction of the virus instead of its constant inflow. We didn't think it's an interesting result, but reading the reviewers comment, we changed our mind to add some description about the results (Fig. S11; L328-334/L314-318.) This effect will be also observed for HR- and ER-inducing R genes.

The occurrence of SHR might potentially be interpreted as a scenario of kin selection when the host plants are reproducing clonally, through selfing, or through severe inbreeding, when all plants of a population in a stand are genetically identical or nearly so, and when the virus is a specialist with a very narrow host range. This places very severe restrictions on when an interpretation of kin selection might be warranted. And even then, I would favor a more generally valid biological interpretation of viewing SHR as a plastic phenotypic outcome that constrains natural selection on immunity.

In the case of CMV in particular the above-mentioned assumptions of when kin selection might potentially be invoked will be extremely rare in natural settings. The virus is a generalist virus that infects more than 1,200 plant species. Chances are slim that a population of one of these host plant species would grow in monoculture in natural settings. However, the occurrence of SHR in one host population would still remove potential sources of virus inoculum from the community of susceptible plants, although community connections also depend on the host ranges of virus-vectoring aphids. SHR would thus influence patterns of selection on resistance in multiple plant populations in this community regardless of whether they are kin of or more distantly related to the infected plant. The occurrence of SHR in one plant species could thereby even constrain selection on resistance to CMV in populations of other plant species.

The descriptions of the model simulations should make more mention of any limitations on the ecological settings to which the simulations may apply.

As our model is not a quantitative one, it is difficult to discuss how close the simulated situation and real plant reproductions are. What we can say still is, that SHR induction can be more adaptive than full susceptibility if reproduction occur locally. Although we have already touched the limitation of our simulations in the previous version of our manuscript, we added some descriptions to explain it further (L410-417/L394-400). We also discuss the situation that SHR induction is less adaptive than full susceptibility, with an aim to make a fair discussion (L471-484/L452-464, already appeared above.)

It is true that CMV is a generalist and may not be an ideal example in this case. However, as many other plant viruses are thought to be specialists (though it is difficult to determine real host range of a virus), and as we aimed to discuss general virus-plant interactions here, we suppose the simulated situation can occur in nature at least in some circumstances. We modified the manuscript to clarify this point (L417-423/L400-407.)

In my opinion, the “inefficient induction of antiviral R-gene-mediated resistance” as in the title, should not be viewed “as a suicide strategy to save the kin from the virus”, but could more appropriately be viewed as phenotypic plasticity that constrains natural selection for resistance or susceptibility.

I agree that the reviewer’s idea is also an attractive one. To discuss further, we need more field surveys to determine many parameters: the virus inflow rate/frequency (as we have discussed above); phenotype variations in different plant-genotype-by-virus-genotype combinations found in real field; aphid preference etc. We hope the reviewer let us leave these tough surveys to future studies.

3) This last point brings me to another aspect and that is how the authors seem to assume that susceptibility to virus infections can only be selected against. Depending on the ecological settings selection may act towards favoring susceptibility. More and more examples of infections having positive fitness consequences depending on the identity of plant and virus and the ecological settings are being published (e.g., Groen et al., PLoS Pathogens, 2016; Xu et al., New Phytologist, 2008). And as mentioned above, also here, lowering the number of infection sources would change patterns of selection for susceptibility just as much as they would change patterns of selection for full immunity.

We agree that in some plant-virus combinations virus infection is beneficial to a host, at least in certain conditions; we agree that such “friendly” interactions are becoming popular in our research area. According to the reviewers comment, we added a simulation for such a friendly situation by decreasing the d_{ni} parameter [i.e., death rate of infected SHR- plants] value lower than d (death rate of uninfected plants) (specifically, 0.08 and 0.1, respectively,) and demonstrated that *SHR*(-) plants are selected very quickly (Fig. S12 and L339-346/L324-331). This result fit the intuition that adaptive genotypes will be selected; we agree that such situation can also occur in nature. However, we consider that it is rather fair to note that the existence of many *R* genes in plant genomes in turn imply that unbeneficial interactions are also general. We added a paragraph to discuss the generality of beneficial and unbeneficial interactions in Discussion (L450-

455/L431-436.).

I agree with the authors that SHR can form an entry point for R-gene adaptation as they mention in the Discussion, but I do not agree that this is necessarily a “stable” entry point. Depending on the ecological settings and the phenotypic and fitness consequences of a fully susceptible interaction, the population may just as well revert from SHR towards full virus susceptibility.

We agree that SHR is not necessarily a stable point; we modified the description (L481-484/L462-464, already appeared above).

Overall, I think the authors have put together an intellectually stimulating manuscript based on a thorough effort of biological experiments and modeling simulations. The manuscript was written very well, which made the authors’ ideas not difficult to understand.

We thank the reviewer again for the critical but constructive comments to our manuscript. We hope that our modifications made the points clear and deepened the discussion.

Reviewer #3 (Remarks to the Author):

The authors Abebe et al present interesting idea on the evolution of plant – virus interaction. They support is with studying infectivity of viral mutants in diverse plant genetic backgrounds and with population level model simulations. There are some minor points to correct and improve.

We thank the reviewer for valuable suggestions to improve our manuscript.

-l61: there seem to be different molecular mechanisms leading to ER, see also <https://onlinelibrary.wiley.com/doi/full/10.1111/pbi.13230>. Also the further explanations (till l66 is maybe not appropriate for the introduction as it describes your opinion. Move it to discussion

We have added the suggested study and some others to expand the examples of ER induction (L68-72/L60-65).

-l136-138: please specify exactly which combinations give HR and which ER, this is also not visible in figure 3 , please provide better images showing necrosis

We thank the reviewer for the comment. We have added a table for summarized results (Table 1 in the new version) and provided Fig. S5 in addition (L163-165/L151-153). We show 7 dpi *A. thaliana* plants in Fig. 3 to explain systemic infection in some combinations. At 7 dpi, systemic necrosis was not occurring for the *R(+)* *A. thaliana* & CMV-CP T45M combination, but systemic necrosis became visible at 14 dpi as shown in Fig. S5.

- l144: Plant RNA viruses accumulate 106–107 copies in each infected cell (e.g., Takebe and Ohtsuki, 1969). – this is very old citation and covers only one pathosystem. At least our data show more than 10-fold differences in different viral pathosystems. Please recheck the generality of the statement

We added more studies to make it general (L178-180/L165-167), though they are somewhat old studies. Unfortunately, we couldn't find recent papers that estimated the number of viral genomes in a cell.

-definition of MOI is well described in results section, thus I propose to modify it in l26 to The average number of viral genomes that establish infection in next cell

We followed this suggestion (L26/L26).

- the manuscript Qu et al is not yet published, thus we cannot evaluate it's relevance for

this paper

The paper is now published. We have added its citation (L392/L377).

-L352-353: From this sentence one understands that it is PCD that is killing the virus. Which is not true, it is a side effect probably aiming at other pathogens. My suggestion is to change it to ' Our simulations imply that SHR response can function as a suicide strategy to save closely related kin from viral infection, contributing to a mechanism for population-level resistance.'

We have added Fig. S4 in Results section: western blot analysis shows that we cannot detect viral CP from necrotic part, suggesting that PCD killed the virus (L155-157/L143-145 and L270-273/L258-260 in Results). We thank the reviewer for raising this important point.

-L358-359: 'The invention of a suicide strategy may be a plausible explanation for this expansion in the number of R genes in land plants.' Please rephrase this part as there are other more direct explanations for the expansion in number of R-genes. For sure they are evolving to cope with multitude of organisms they are in contact with

We agree that the number of R genes reflect the number of organisms in contact with. We have added the description; because our point is that such a resistance mechanism is not functional in organisms other than land plants, we added some words to explain the point (L425-428/L408-411).

-L364: Considering that a suicide strategy incorporating SHR requires cell death,... You have not proved that cell death is required, only that the plant needs to die. The plants die also without SHR, in 'normal' compatible interaction. Please rephrase

In the case of compatible interaction with viruses, plants usually do not die that quickly, and this makes a big difference from SHR in whether the infected plant serves as infection source or not. We have added a western blot analysis for CP (Fig. S4), and showed that viral particles are degraded by cell death. We have added descriptions about this experiment (L155-157/L143-145).

- authors should consider the rewriting of the abstract as the message of their achievements was for me not clear when reading the abstract before the manuscript

Due to the strict word number limitation, we needed to skip the details. However, we have tried to improve it. We hope that it is clearer now.

We again thank the reviewer for the comments that was important to clarify our points and improve the manuscript.

Additional modifications:

In addition to the modifications to answer reviewers' comments, we corrected a mistake in virus name in Introduction: the study by Jones and Vincent (2018) was on potato virus Y (PVY), but not on potato virus X (PVX). We are sorry for this mistake. We have also made the following modifications that will not affect the review process:

- 1) Data availability statement and material availability statement were added (L786-796/L766-776).
- 2) Use of a computing resource was added to Acknowledgements (L767-769/L747-749).

Reviewers' comments:

Reviewer #2 (Remarks to the Author):

The authors have done a lot of work to address to comments that the two other reviewers and I raised, and in my opinion the manuscript has become more balanced and stronger. I am excited about this manuscript, and I suggest only minor edits to the text. These edits would remove unwanted hints remaining in the text here and there that evolution could have a purpose. They would also cause the text to make fewer references to the controversial topic of kin selection, and I feel like this could make the manuscript land better with a wide audience, which the authors' ideas deserve.

Please find my suggested edits below.

Title: "saving" instead of "to save" to avoid implication of directionality in evolution

Abstract

Line 29: revert back to "a" instead of "its"

Introduction

Line 42: "to" instead of "for"

Line 51: remove closing parenthesis after "programmed"?

Line 77: remove "either"

Line 78: "loci" instead of "locus"

Line 88: change "; involvement of common" to "; and the involvement of a common"

Line 91: add "the" before "NtTPN1"

Results

Line 155: "is" instead of "in"

Line 171: add comma after "thus SHR", and remove comma after "ER in"

Line 240: change "an alternative story is necessary." to something like "the existence of SHR would need to be explained by one or more alternative mechanisms."?

Lines 241: change the sentence starting with "The alternative story ..." to "An alternative mechanism may be found in population-level antiviral resistance."?

Line 253: add "having" between "without" and "the"

Line 273: change "amount" to "amounts"

Line 339: "has" instead of "have"

Line 340: "effects on" instead of "effect to"

Line 342: add "a" after "giving"

Discussion

Line 409: "that saves" instead of "to save"

Line 413: change "to prove that kin selection is really at work in nature" to something like "to establish if SHR could have positive fitness consequences for neighbouring plants in nature"?

Line 441: add "could be beneficial" after "population resistance"?

Line 466: change "field" to "fields"

Line 478: change "helps" to something like "promotes"

Line 479: change "and" to "or"

Line 482: change "helps SHR serve" to "promotes SHR as"

Line 494: I feel like a concluding statement might be missing here, the Discussion section seems to end rather abruptly here.

Figures

Line 933: change "to protect" to "that may protect"

Line 1012: "does not" instead of "do not"

Reviewer #3 (Remarks to the Author):

-L352-353: From this sentence one understands that it is PCD that is killing the virus. Which is not true, it is a side effect probably aiming at other pathogens. My suggestion is to change it to ' Our simulations imply that SHR response can function as a suicide strategy to save closely related kin from viral infection, contributing to a mechanism for population-level resistance.'

Authors response: We have added Fig. S4 in Results section: western blot analysis shows that we cannot detect viral CP from necrotic part, suggesting that PCD killed the virus (L155-157/L143- 145 and L270-273/L258-260 in Results). We thank the reviewer for raising this important point.

Reviewer 2nd response: Western blot analysis cannot confirm that PCD is blocking the virus. The only way to do it is to have virus tagged with fluorescent protein and visualise its spread under confocal microscope. There are several papers out supporting this information , thus you would need more evidence to claim that your pathosystem works differently. This is reviewed in Künstler et al., 2016, studied in Farkas et al., 1960; Király et al., 2008; Hafez et al., 2012, Lukan et al 2018
Please take these into account and rephrase the statement.

Reviewer #2 (Remarks to the Author):

The authors have done a lot of work to address to comments that the two other reviewers and I raised, and in my opinion the manuscript has become more balanced and stronger. I am excited about this manuscript, and I suggest only minor edits to the text. These edits would remove unwanted hints remaining in the text here and there that evolution could have a purpose. They would also cause the text to make fewer references to the controversial topic of kin selection, and I feel like this could make the manuscript land better with a wide audience, which the authors' ideas deserve.

We thank the reviewer for the kind and helpful suggestions for appropriate wordings. We followed all the wording suggestions, and added a missing sentence to conclude the last paragraph in Discussion as below (L484-487/L482-484):

“These insights into possible trajectories of antiviral *R* gene evolution will benefit future studies on evolution histories of plant–pathogen interactions, as well as studies on agricultural application of *R* genes.”

We would like to express our sincere thanks to the reviewer for his/her great helps in the current and previous rounds of revision.

Please find my suggested edits below.

Title: “saving” instead of “to save” to avoid implication of directionality in evolution

Abstract

Line 29: revert back to “a” instead of “its”

Introduction

Line 42: “to” instead of “for”

Line 51: remove closing parenthesis after “programmed”?

Line 77: remove “either”

Line 78: “loci” instead of “locus”

Line 88: change “; involvement of common” to “; and the involvement of a common”

Line 91: add “the” before “NtTPN1”

Results

Line 155: “is” instead of “in”

Line 171: add comma after “thus SHR”, and remove comma after “ER in”

Line 240: change “an alternative story is necessary.” to something like “the existence of SHR would need to be explained by one or more alternative mechanisms.”?

Lines 241: change the sentence starting with “The alternative story ...” to “An alternative mechanism may be found in population-level antiviral resistance.”?

Line 253: add “having” between “without” and “the”

Line 273: change “amount” to “amounts”

Line 339: “has” instead of “have”

Line 340: “effects on” instead of “effect to”

Line 342: add “a” after “giving”

Discussion

Line 409: “that saves” instead of “to save”

Line 413: change “to prove that kin selection is really at work in nature” to something like “to establish if SHR could have positive fitness consequences for neighbouring plants in nature”?

Line 441: add “could be beneficial” after “population resistance”?

Line 466: change “field” to “fields”

Line 478: change “helps” to something like “promotes”

Line 479: change “and” to “or”

Line 482: change “helps SHR serve” to “promotes SHR as”

Line 494: I feel like a concluding statement might be missing here, the Discussion section seems to end rather abruptly here.

Figures

Line 933: change “to protect” to “that may protect”

Line 1012: “does not” instead of do not”

Reviewer #3 (Remarks to the Author):

We sincerely thank the reviewer for reviewing the revised version of the manuscript.

-L352-353: From this sentence one understands that it is PCD that is killing the virus. Which is not true, it is a side effect probably aiming at other pathogens. My suggestion is to change it to ' Our simulations imply that SHR response can function as a suicide strategy to save closely related kin from viral infection, contributing to a mechanism for population-level resistance.'

Authors response: We have added Fig. S4 in Results section: western blot analysis shows that we cannot detect viral CP from necrotic part, suggesting that PCD killed the virus (L155-157/L143- 145 and L270-273/L258-260 in Results). We thank the reviewer for raising this important point.

Reviewer 2nd response: Western blot analysis cannot confirm that PCD is blocking the virus. The only way to do it is to have virus tagged with fluorescent protein and visualise its spread under confocal microscope. There are several papers out supporting this information , thus you would need more evidence to claim that your pathosystem works differently. This is reviewed in Künstler et al., 2016, studied in Farkas et al., 1960; Király et al., 2008; Hafez et al., 2012, Lukan et al 2018

Please take these into account and rephrase the statement.

We are sorry that we have been misleading the reviewer for this point. We do not mean that necrosis stops the virus as an individual-level resistance. Below I explain some important points and the modifications made to the revised manuscript for clarity:

1) As described in L51-57/L51-57, we also consider that necrosis itself does not stop the virus in the infected tissues. We have also observed the accumulation of viral protein (CP in our case) in the marginal region of necrotic tissues but not within the necrotic tissues, by press-blot analysis in our previous study (Sekine et al., 2008) and in the current study, and thus added an image as Fig. S4b (I have also put the image here). We have added some sentences to describe our press-blot results (L145-149/L145-149). Our press-blot experiment

is a good alternative of the fluorescent-protein-based experiment suggested by the reviewer; we couldn't do the exact experiment because, in our system, necrotic tissues exhibit strong autofluorescence, which is difficult to distinguish from virus-derived fluorescence.

2) As described in L339-344/L338-343, the current study detected a significant decrease in MOI before necrosis induction; this result supports the existence of PCD-independent virus resistance in the inoculated leaves. To further clarify our point, we have added some references (suggested by the reviewer and additional ones: L340-342/L339-341).

3) In L419-423/L417-422, we again touch that PCD induction and viral containment occur via independent pathways; we have added an appropriate reference suggested by the reviewer.

4) As we did know the abovementioned viral accumulation in the marginal region, we carefully sampled the inside necrotic tissue for the Western blot analysis shown in Fig. S4a. We literally sampled "necrotic tissues"; by this way, we here clearly demonstrate that viral particles are degraded within the necrotic tissues. Though necrosis itself is dispensable for stopping the virus in the infected leaves, virus degradation demonstrated here can stop plant-to-plant spread of the virus. In other words, necrosis can function as population-level resistance, but not as individual-level resistance, both in HR and SHR. We discuss this point in L424-428/L422-426. We added a description to touch ER to explain that necrosis is required only after virus accumulated to some extent L430-431/L428-429).

We believe that we could further clarify our points by the above modifications, and could make it clear that the authors do not have contradictory results with previous studies. We thank the reviewer for his/her helpful comments.

REVIEWERS' COMMENTS:

Reviewer #3 (Remarks to the Author):

the authors have addressed the remaining issues, thus I fully support publication of the manuscript